# Robust Classification with Noisy Labels Based on Posterior Learning

## Abstract

Deep learning has shown robustness to label noise under specific assumptions, yet its performance under extremely high noise rates remains a significant challenge. In this paper, we theoretically demonstrate under what conditions models estimating the posterior probability can achieve high classification accuracy in the presence of extremely strong instance-dependent label noise without performing loss correction approaches. To estimate the noisy posterior, we propose a class of objective functions derived from the variational representation of the $f$-divergence. Furthermore, we propose two correction methods to achieve robustness when the algorithm is not intrinsically robust to label noise: one method is implemented during the training process, and the other is performed during inference. Finally, we show the validity of our theoretical results and the effectiveness of the proposed methods on synthetic and real-world label noise settings.

## 1 Introduction

The success of large deep neural networks is highly dependent on the availability of large labeled datasets. However, the labeling process is often expensive and sometimes imprecise, whether it is done by human operators or by automatic labeling tools. On average, datasets contain from $8\%$ to $38.5\%$ of samples that are corrupted with label noise Song et al. (2022); Xiao et al. (2015); Li et al. (2017); Lee et al. (2018); Song et al. (2019).

For supervised classification tasks, different strategies focused either on the architecture and training strategy development or on the objective function design. Although effective objective functions have been proposed Hui & Belkin (2020); Dong et al. (2019); Blondel et al. (2019); Novello & Tonello (2024), the cross-entropy (CE) is still the most frequent approach. Meanwhile, multiple studies showed that the standard CE minimization is a poor choice for classification in the presence of label noise, mainly due to its gradients and the data memorization phenomenon Ghosh et al. (2017); Zhang & Sabuncu (2018); Liu et al. (2020). Consequently, a significant body of research has focused on substituting the CE with objective functions that are either intrinsically robust to label noise or robust after a correction operation Patrini et al. (2017); Bae et al. (2024); Liu & Guo (2020); Wei & Liu (2021); Ma et al. (2020); Ye et al. (2023). These methods lead to statistically consistent classifiers, theoretically ensuring the convergence to the optimal classification performed on the clean data.

However, all the algorithms that have been proposed as intrinsically robust to label noise (i.e., robust without performing correction techniques) rely on the assumption that the noise rate is smaller than $\frac{K-1}{K}$, where $K$ is the number of classes. We investigate what happens when this assumption is violated, a state we refer to as *extreme label noise* conditions. So far, correction-based approaches are the only ones that can handle such a violation, but they require the precise knowledge of the noise transition probabilities. We prove theoretically and demonstrate numerically that in some scenarios (e.g., symmetric label noise) it is possible to solve classification under extreme label noise without applying any correction operation, but estimating a sample's class by minimizing the noisy posterior probability. Combining these new results with those in Zhu et al. (2024), we observe that tackling label noise problems with deep learning algorithms that estimate the noisy posterior density is an effective choice, as in a wide variety of scenarios they are already intrinsically robust to label noise. However, although at optimal convergence any loss function estimating the posterior would be suitable, in practice the loss function choice is fundamental. For instance, it has already been demonstrated that the commonly used CE is not appropriate for label noise scenarios Liu et al.

(2020). Therefore, we propose an objective function designed from the variational representation of the $f$-divergence, which can estimate the posterior using any $f$-divergence. We refer to it as $f$-divergence-based Noisy Posterior Learning ($f$-NPL). We observe that $f$-NPL can be seen as a specific case of active passive losses (APLs) Ma et al. (2020), thus mitigating the "underfitting" problem. In some cases where the noisy posterior is not intrinsically robust to label noise, we propose two correction techniques to make $f$-NPL robust. The former is applied during the training phase to learn a neural network equal to the one trained with the clean dataset. The latter is performed during the test phase to correct the posterior probability estimate, making it robust to label noise. Finally, experimental results demonstrate the validity of our theoretical findings about extreme label noise rates and the effectiveness of the proposed correction approaches on synthetic and real-world label noise settings, outperforming other correction-based and APL-like methods.

Synthetically, the key contributions of this paper are:

- We prove that under certain assumptions, in extreme label noise conditions, the true label can be estimated by minimizing the noisy posterior, without requiring loss correction.
- We propose $f$-NPL, and we provide novel approaches to correct either the objective function of $f$-NPL (during training) or the posterior estimator (during test), to achieve robustness in scenarios where $f$-NPL is not intrinsically robust to label noise.

## 2  RELATED WORK

In this section, we provide a summary of the existing approaches designing objective functions for classification in the presence of label noise, while we defer to Appendix C for additional techniques.

**Objective function correction**    These methods rely on the idea of modifying the objective function to improve the label noise robustness of the classifier. Typically, these algorithms require the knowledge of the matrix of transition probabilities from true labels to noisy labels (i.e., transition matrix). When the transition matrix is not known, it can be estimated, as studied in Patrini et al. (2017); Yao et al. (2020); Li et al. (2021); Zhang et al. (2021); Cheng et al. (2022). In Natarajan et al. (2013), the authors propose a weighted loss function for binary classification in the presence of class-conditional noise. In Liu & Tao (2015), the authors utilize the transition matrix to employ reweighting, which utilizes importance sampling to ensure robustness. Forward and Backward Patrini et al. (2017) are two algorithms for loss correction given the transition matrix, which is estimated finding the dataset anchor points. In Bae et al. (2024), the authors propose a resampling technique that works better than reweighting in the label noise scenario.

**Robust objective functions**    These algorithms utilize objective functions inherently robust to label noise. In Menon et al. (2015), the authors demonstrate the robustness of deep learning algorithms for binary classification under symmetric label noise. In Ghosh et al. (2017), the authors prove the robustness of symmetric objective functions. In particular, they show that the CE is not symmetric, while proving that the mean absolute error (MAE) is a robust loss. In Ma et al. (2020), the authors prove that all the objective functions can be made robust to label noise with a normalization. However, they show that robust losses can suffer from an underfitting issue. Therefore, they propose a class of objective functions, referred to as active passive losses (APLs), that mitigate the underfitting problem. In Ye et al. (2023), the authors propose a specific class of APLs, referred to as active negative loss functions (ANLs), that, instead of obtaining the passive losses based on MAE as in Ma et al. (2020), use negative loss functions based on complementary label learning Ishida et al. (2017). In Zhou et al. (2023), the authors propose a class of loss functions robust to label noise that extend symmetric losses, while stressing the urgency of designing non-symmetric objective functions robust to label noise. In Zhu et al. (2024), the authors prove the robustness of deep learning algorithms to instance-dependent noise when the transition matrix is strictly diagonally dominant.

## 3  ROBUST CLASSIFICATION WITH LABEL NOISE

In a supervised classification problem, a classifier is learned using a clean dataset $\{(\mathbf{x}_1, y_{\mathbf{x}_1}), \ldots, (\mathbf{x}_N, y_{\mathbf{x}_N})\} \equiv \mathcal{D}$ drawn i.i.d. from $X \times Y$. Differently, in the weakly-supervised

scenario of classification in the presence of label noise, we can only access a noisy dataset $\{(\mathbf{x}_1, \tilde{y}_{\mathbf{x}_1}), \ldots, (\mathbf{x}_N, \tilde{y}_{\mathbf{x}_N})\} \equiv \mathcal{D}_\eta$ drawn from $X \times Y_\eta$. We denote as $D(\cdot)$ the classifier. In the presence of label noise, instead of estimating $p_{Y|X}$, a classifier can only estimate $p_{Y_\eta|X}$. In general, for instance-dependent label noise, the noisy and clean posterior probabilities are related through a noise transition matrix-valued function $T : \mathcal{X} \to \{M \in \mathbb{R}^{K \times K} : M \text{ is row stochastic}\}$, also known as the noise transition matrix, which has the $(i, j)$-th entry defined as

$$[T(\mathbf{x})]_{i,j} = \mathbb{P}\left(Y_\eta = j | Y = i, X = \mathbf{x}\right), \tag{1}$$

thus implying that $p_{Y_\eta|X}(y_{\mathbf{x}}|\mathbf{x}) = T(\mathbf{x})^T p_{Y|X}(y_{\mathbf{x}}|\mathbf{x})$. The noise rate $\mathbb{P}(Y_\eta \neq Y)$ is referred to as $\eta$.

A classification algorithm is *noise tolerant* (i.e., robust to label noise) when the classifier learned on noisy data has the same probability of correct classification as the classifier learned on clean data Manwani & Sastry (2013), i.e.,

$$\mathbb{P}(pred \circ D^\diamond(\mathbf{x}) = y_{\mathbf{x}}) = \mathbb{P}(pred \circ D_\eta^\diamond(\mathbf{x}) = y_{\mathbf{x}}), \tag{2}$$

where $pred$ indicates the function used to predict the class (e.g., argmax), $D^\diamond$ is the neural network trained on the clean dataset, and $D_\eta^\diamond$ is the neural network trained in presence of label noise. Usually, the label noise robustness is proved by demonstrating that a certain objective function is symmetric Ghosh et al. (2017), meaning that the sum of the losses computed over all the classes is constant. The objective function symmetry leads to the condition $D^\diamond(\mathbf{x}) = D_\eta^\diamond(\mathbf{x})$ Ghosh et al. (2017); Ma et al. (2020), trivially proving the label noise robustness. However, that is only a sufficient condition for equation 2 to be true. Therefore, there can be objective functions that are robust to label noise but for which $D^\diamond(\mathbf{x}) \neq D_\eta^\diamond(\mathbf{x})$. This is exactly the case of the techniques estimating the noisy posterior, such as $f$-NPL (proposed in Sec. 4).

All the results on intrinsic label noise robustness of classification algorithms guarantee, at best, the robustness up to a noise rate $\eta < \frac{K-1}{K}$. Zhu et al. (2024) proved that learning algorithms that estimate the one-hot posterior $p_{Y|X}$ are robust to instance-dependent label noise under the assumption that the transition matrix is strictly diagonally dominant, which can be verified when $\eta < \frac{K-1}{K} \triangleq T_h$. In this section we tackle the question: *what happens when $\eta > T_h$?* First, we prove in Theorem 3.1 that, for the general case of instance-dependent label noise, when the transition matrix has the elements of the main diagonal which are smaller than any other element in their row (we refer to it as anti-diagonally dominant), we can estimate the true label by minimizing the noisy posterior. This allows to correctly estimate the label without performing any loss correction technique, that would be more computationally demanding. As a byproduct, we prove that in some scenarios with $\eta > T_h$, it is possible to estimate the true label with the only knowledge that the noise rate exceeds $T_h$, without knowing the exact values of the noise transition probabilities. One prominent example is symmetric label noise, analyzed in Corollary 3.2, where the true label transitions to any other label with equal probability.

**Theorem 3.1.** *Let $p_{Y|X}(\cdot|\mathbf{x}) \in \{\mathbf{e}_1, \ldots, \mathbf{e}_K\}$ be a one-hot vector, and assume the diagonal elements of $T(\mathbf{x})$ minimize their rows, then*

$$\arg\max_y p_{Y|X}(y|\mathbf{x}) = \arg\min_y p_{Y_\eta|X}(y|\mathbf{x}). \tag{3}$$

**Corollary 3.2.** *For symmetric label noise, when $\eta > \frac{K-1}{K}$, the class minimizing the noisy posterior coincides with the class predicted by the optimal Bayes classifier in the absence of label noise.*

Theorem 3.1 and Corollary 3.2 show that deep learning algorithms that estimate the posterior probability density are not only robust when the main diagonal of $T(\mathbf{x})$ is dominant (showed in Zhu et al. (2024)), but that, when the transition matrix is anti-diagonally dominant, the true label can be predicted by computing the argmin of the noisy posterior (Theorem 3.1). Previous work was only able to theoretically achieve robustness for extremely high label noise by employing correction approaches that require estimating the entire transition matrix. Meanwhile, Corollary 3.2 proves that when the noise rate exceeds $T_h$, the correct class can be obtained without the exact knowledge of the noise transition probabilities, by minimizing the estimated noisy posterior.

Given these results, one could opt for using a foundation model to extract the dataset features and then solve classification with simple standard machine learning techniques, as proposed in Zhu et al. (2024). However, this procedure has three main drawbacks: 1) there are no foundation models for

Figure 1: Proposed framework in the presence of label noise. The framework is robust to instance-dependent label noise when the transition matrix is strictly diagonally dominant (yellow). For instance-dependent label noise, when the transition matrix is anti-diagonally dominant, the true label can be obtained by minimizing the noisy posterior (pink). For some other scenarios in which the previous assumptions do not hold, to achieve robustness with label noise, the objective function correction (green) is performed during training to obtain the clean estimate of the posterior as the output of the neural network. Alternatively, the posterior correction (blue) is implemented during the test phase by correcting the noisy posterior estimate. The dashed arrows indicate the model update through backpropagation.

any type of signal dataset (such as decoding problems for communications engineering); 2) the theoretical robustness has been guaranteed under assumptions on the main diagonal of $T(\mathbf{x})$, but there is no guarantee when such assumptions are not met; 3) this theoretical analysis does not take into account more practical training considerations, such as the problem of "underfitting" Ma et al. (2020). Therefore, we propose the usage of an $f$-divergence-based class of objective functions that mitigates some of these limitations.

## 4 ROBUST $f$-DIVERGENCE POSTERIOR-BASED CLASSIFICATION

### 4.1 $f$-DIVERGENCE

Given a domain $\mathcal{X}$ and two probability density functions $p(\mathbf{x})$, $q(\mathbf{x})$ on this domain, the $f$-divergence is defined as Ali & Silvey (1966); Csiszár (1967)

$$D_f(p||q) = \int_{\mathcal{X}} q(\mathbf{x}) f\left(\frac{p(\mathbf{x})}{q(\mathbf{x})}\right) d\mathbf{x}, \tag{4}$$

where $p \ll q$ (i.e., $p$ is absolutely continuous with respect to $q$) and where the *generator function* $f : \mathbb{R}_+ \longrightarrow \mathbb{R}$ is a convex, lower-semicontinuous function such that $f(1) = 0$. The variational representation of the $f$-divergence Nguyen et al. (2010) reads as

$$D_f(p||q) = \sup_{D:\mathcal{X}\to\mathbb{R}} \left\{ \mathbb{E}_p\left[D(\mathbf{x})\right] - \mathbb{E}_q\left[f^*(D(\mathbf{x}))\right] \right\}. \tag{5}$$

where $T$ is a parametric function (e.g., a neural network) and $f^*$ denotes the *Fenchel conjugate* of $f$ and is defined as $f^*(t) \triangleq \sup_{u \in dom_f} \{ut - f(u)\}$, with $dom_f$ being the domain of the function $f$. The supremum in equation 5 is attained for $T^\diamond(\mathbf{x}) = f'(p(\mathbf{x})/q(\mathbf{x}))$, with $f'$ first derivative of $f$.

### 4.2 DENSITY-RATIO BASED CLASSIFICATION WITH $f$-DIVERGENCE

The idea is that we want to benefit from the theoretical results of Sec. 3 and Zhu et al. (2024) by training a neural network that estimates the noisy posterior. Trivially, this can be achieved by minimizing the standard CE loss. However, previous work showed that this choice has multiple practical problems, mainly related to the gradients of the CE Liu et al. (2020) and to the absence of a "passive" term that mitigates the problem of "under learning" on some hard classes Ma et al. (2020). We propose to design the objective function as the variational representation of the $f$-divergence between $p_{XY_\eta}$ and $p_X$, tackling part of these problems. Let

$$\mathcal{J}_f(D) = \mathbb{E}_{XY_\eta}\left[D(\mathbf{x})\mathbf{1}_K(y_\mathbf{x})\right] - \mathbb{E}_X\left[\sum_{i=1}^K f^*\left(D(\mathbf{x}, i)\right)\right], \tag{6}$$

where $D(\mathbf{x}) = [D(\mathbf{x}, 1), \ldots, D(\mathbf{x}, K)]$, with $D(\mathbf{x}, i)$ $i$-th component of the neural network's output $D(\mathbf{x})$, and $\mathbf{1}_K(y_\mathbf{x})$ is the one-hot encoded label $y_\mathbf{x}$. Then, at convergence, the learned neural network is a function of the density-ratio $p_{Y_\eta|X} = p_{XY_\eta}/p_X$ Nguyen et al. (2010). Therefore, it is possible to express the posterior estimate as a density-ratio, and to obtain the class estimated by maximizing the noisy posterior as

$$\hat{y}_\mathbf{x} = \arg\max_{y_\mathbf{x} \in \mathcal{A}_y} \hat{p}_{Y_\eta|X}(y_\mathbf{x}|\mathbf{x}) = \arg\max_{y_\mathbf{x} \in \mathcal{A}_y} (f^*)'(D^\diamond(\mathbf{x})), \tag{7}$$

where $D^\diamond(\cdot)$ is the optimal neural network trained by maximizing equation 6. The same variational representation-based approach has been used in Novello & Tonello (2024) for supervised classification. We report the relationship between equation 6 and empirical risk minimization (ERM) in Appendix A.1. In this section, we show that this class of objective functions has many benefits for classification with label noise.

Since this approach performs classification by estimating the noisy posterior, it gains all the robustness results discussed in Section 3. Furthermore, this class of objective functions resembles the active passive losses (APLs), which benefit from the presence of a passive loss mitigating the "underfitting" issue Ma et al. (2020). In fact, the first expectation $\mathbb{E}_{XY}$ in equation 6 is affected only by the neural network's prediction corresponding to the label, while the second expectation $\mathbb{E}_X$ is impacted by the neural network's predictions corresponding to classes different from the label. In contrast to the explicit APL-based objective function design in Ma et al. (2020); Ye et al. (2023), where the active and passive terms are unrelated, the variational formulation of $f$-NPL leads to an APL-like objective function that synchronizes the active and passive terms by implicitly considering their interdependency, allowing us to estimate the noisy posterior. In Appendix C, we provide more details and we provide additional comparisons with other related work.

In Sections 4.3 and 4.4, we develop ad-hoc correction methods for $f$-NPL which work also when the assumption on the main diagonal of the transition matrix is relaxed. We will restrict our analysis to class-conditional label noise (i.e., $\mathbb{P}(Y_\eta|Y, X) = \mathbb{P}(Y_\eta|Y)$), for which the noisy label is generated as

$$\tilde{y}_\mathbf{x} = \begin{cases} y_\mathbf{x} & \text{with probability } (1 - \eta_{y_\mathbf{x}}) \\ j, j \in [K], j \neq y_\mathbf{x} & \text{with probability } \eta_{y_\mathbf{x}j} \end{cases}, \tag{8}$$

where $\eta_{y_\mathbf{x}j}$ represents the transition probability from the true label $y_\mathbf{x}$ to the noisy label $j$, i.e., $\eta_{y_\mathbf{x}j} = \mathbb{P}(Y_\eta = j|Y = y_\mathbf{x})$, and $j \in [K]$ is a concise notation for $j \in \{1, \ldots, K\}$. $\eta_{y_\mathbf{x}} = \sum_{j \neq y_\mathbf{x}} \eta_{y_\mathbf{x}j}$ is the noise rate. The correction techniques rely on the hypothesis of having the transition probabilities $\eta_{y_\mathbf{x}j}$. When the transition probabilities are unknown, they can be estimated, as outlined in Sec. 2.

### 4.3 OBJECTIVE FUNCTION CORRECTION

In this section, we present an objective function correction approach performed during training, designed to ensure convergence to the neural network that would be learned using the clean dataset. We first study the binary classification case and then extend it to multi-class classification.

#### 4.3.1 BINARY OBJECTIVE FUNCTION CORRECTION

Let $Y = \{0, 1\}$ be the labels set. Define the following quantities: $e_0 \triangleq \mathbb{P}(Y_\eta = 0|Y = 1)$, $e_1 \triangleq \mathbb{P}(Y_\eta = 1|Y = 0)$ for simplicity in the notation. In the following, we always assume $e_0 + e_1 < 1$. Theorem 4.1 shows the effect of label noise on the class of objective functions in equation 6.

**Theorem 4.1.** *For binary classification, the relationship between the value of the objective function in the presence ($\mathcal{J}_f^\eta(D)$) and absence ($\mathcal{J}_f(D)$) of label noise, given the same parametric function $D$, is*

$$\mathcal{J}_f^\eta(D) = (1 - e_0 - e_1)\mathcal{J}_f(D) + B_f(D), \tag{9}$$

*where $B_f(D) \triangleq \mathbb{E}_X \left[ e_0 D(\mathbf{x}, 0) + e_1 D(\mathbf{x}, 1) - (e_0 + e_1) \sum_{i=0}^{1} f^*(D(\mathbf{x}, i)) \right]$ is a bias term.*

In corollary 4.2, we show how to perform the objective function correction to remove the effect of label noise.

**Corollary 4.2.** *Let us assume the label noise transition probabilities are correctly estimated. Define*

$$\mathcal{J}_f^{\eta,C}(D) \triangleq \mathcal{J}_f^{\eta}(D) - \hat{B}_f(D), \tag{10}$$

*where $\hat{B}_f(D)$ is the estimated bias term. Then,*

$$D^{\diamond} = \arg\max_D \mathcal{J}_f(D) = \arg\max_D \mathcal{J}_f^{\eta,C}(D). \tag{11}$$

Corollary 4.2 directly follows from Theorem 4.1, since the bias estimate $\hat{B}_f(D)$ is accurate (i.e., $\hat{B}_f(D) = B_f(D)$) when the transition matrix is correctly estimated or known. Then, the maximization of $(1 - e_0 - e_1)\mathcal{J}_f(D)$ over $D$ is equivalent to the maximization of $\mathcal{J}_f(D)$.

### 4.3.2 MULTI-CLASS OBJECTIVE FUNCTION CORRECTION

Let us first define the notation for the multi-class classification case with asymmetric uniform off-diagonal label noise Liu & Guo (2020); Wei & Liu (2021): $e_j \triangleq P(Y_\eta = j|Y = i) = \eta_{ij} \,\forall i \neq j$. Assume $\sum_{j \neq i} e_j < 1$. Theorem 4.3 extends Theorem 4.1 for the multi-class case.

**Theorem 4.3.** *For multi-class asymmetric uniform off-diagonal label noise, the relationship between the value of the objective function in the presence ($\mathcal{J}_f^{\eta}(D)$) and absence ($\mathcal{J}_f(D)$) of label noise, given the same parametric function D, is*

$$\mathcal{J}_f^{\eta}(D) = \Big(1 - \sum_{j=1}^{K} e_j\Big)\mathcal{J}_f(D) + B_f(D), \tag{12}$$

*where $B_f(D) \triangleq \mathbb{E}_X\Big[\sum_{j=1}^{K}\Big(e_j D(\mathbf{x}, j) - \Big(\sum_{i=1}^{K} e_i\Big) f^*(D(\mathbf{x}, j))\Big)\Big]$.*

Corollary 4.2 holds true also for the multi-class extension, for the same motivation provided in the binary scenario.

*Sparse label noise* is another model considered in the literature Wei & Liu (2021); Wei et al. (2022), where each original label has a unique counterpart it can be flipped to, meaning that this noise models pairwise label swaps. Assuming $K$ even number, in this scenario there are $K/2$ disjoint pairs of classes $(i_c, j_c)$ with $c \in \left[\frac{K}{2}\right]$ and $i_c < j_c$ with $\eta_{j_c i_c} = e_0$ and $\eta_{i_c j_c} = e_1$. The multi-class classification problem with sparse label noise can be treated as $K/2$ separate binary problems. Additional details, including the description of how to make $f$-NPL robust to sparse label noise, are reported in Appendix B.7.

## 4.4 POSTERIOR ESTIMATOR CORRECTION

In this section, we present an alternative correction procedure to remove the effect of label noise during the test phase, acting on the posterior estimator obtained by training the neural network with the noisy dataset. Let $\hat{p}_{Y|X}$ and $\hat{p}_{Y_\eta|X}$ be the posterior estimators obtained with the clean and noisy datasets, respectively. In general,

$$\hat{y}_{\mathbf{x}} = \arg\max_y \hat{p}_{Y|X}(y|\mathbf{x}) \neq \arg\max_y \hat{p}_{Y_\eta|X}(y|\mathbf{x}) = \hat{y}_{\mathbf{x}}^{\eta}. \tag{13}$$

First, we study the relationship between $\hat{p}_{Y|X}$ and $\hat{p}_{Y_\eta|X}$ by making explicit the effect of label noise in the expression of $\hat{p}_{Y_\eta|X}$. Then, we show how to correct the posterior estimate to make it robust to label noise. From now, we refer to $D^{\diamond}$ and $D_{\eta}^{\diamond}$ as the neural network learned without and with label noise, respectively, as we need this differentiation to relate the two posterior probability densities.

### 4.4.1 BINARY POSTERIOR CORRECTION

Theorem 4.4 describes the relationship between $f$-NPL's posterior estimator in the presence and absence of label noise. In Corollary 4.5, we propose to correct the estimate of the posterior to remove the effect of label noise.

**Theorem 4.4.** *(see proof of Lemma 7 in Natarajan et al. (2013)) For the binary classification case, the posterior estimator in the presence and absence of label noise are related as*

$$\hat{p}_{Y_\eta|X}(i|\mathbf{x}) = (f^*)'(D_\eta^\diamond(\mathbf{x}, i)) = (1 - e_0 - e_1)\hat{p}_{Y|X}(i|\mathbf{x}) + e_i, \quad \forall i \in \{0, 1\}. \quad (14)$$

**Corollary 4.5.** *Let us assume the transition probabilities are correctly estimated. Define* $\hat{p}_{Y_\eta|X}^C(i|\mathbf{x}) \triangleq \hat{p}_{Y_\eta|X}(i|\mathbf{x}) - \hat{e}_i$. *Then,*

$$\hat{y}_\mathbf{x} = \arg\max_{y \in \mathcal{A}_y} \hat{p}_{Y|X}(y|\mathbf{x}) = \arg\max_{y \in \mathcal{A}_y} \hat{p}_{Y_\eta|X}^C(y|\mathbf{x}). \quad (15)$$

Corollary 4.5 follows from the fact that the estimate of the class is computed by maximizing $\hat{p}_{Y_\eta|X}^C(y_\mathbf{x}|\mathbf{x})$ w.r.t. the class element. Therefore, the multiplication by the positive constant does not affect the argmax of the posterior, making it possible to solve the classification problem using $\hat{p}_{Y_\eta|X}^C(y_\mathbf{x}|\mathbf{x})$ in equation 7.

### 4.4.2 MULTI-CLASS POSTERIOR CORRECTION

Theorem 4.6 extends Theorem 4.4 for the case of asymmetric uniform off-diagonal label noise. Corollary 4.5 holds also for the multi-class case analyzed by Theorem 4.6.

**Theorem 4.6.** *For multi-class asymmetric uniform off-diagonal label noise, the relationship between the posterior estimator in the presence and absence of label noise is*

$$\hat{p}_{Y_\eta|X}(i|\mathbf{x}) = (f^*)'(D_\eta^\diamond(\mathbf{x}, i)) = \left(1 - \sum_{j=1}^K e_j\right)\hat{p}_{Y|X}(i|\mathbf{x}) + e_i, \quad \forall i \in \{1, \ldots, K\}. \quad (16)$$

A key distinction between the posterior correction and the objective function correction is at what stage of the algorithm they are applied. Specifically, Corollary 4.2 removes the bias during training, ensuring that maximizing the objective function is equivalent under both noisy and clean conditions. Therefore, the neural network learned in the noisy setting is equal to the one trained on the clean data. In contrast, posterior estimator correction in Corollary 4.5 is performed during the test phase. Although the neural network trained with noisy labels differs from its counterpart trained on clean data, subtracting the bias during posterior correction leads to a maximization of the corrected posterior (w.r.t. the class $y_\mathbf{x}$) in the noisy setting that is equivalent to the maximization of the posterior in the clean scenario.

**Convergence analysis** It is possible to theoretically quantify the bias between the true posterior, the posterior estimator attained maximizing $\mathcal{J}_f^\eta(D)$, and the value of such an estimator during training. Theorems B.1 and B.2 in Appendix B show that the bias during training depends on the chosen $f$-divergence. Furthermore, they provide a characterization of the bias as a function of the $f$-divergence employed, demonstrating that it depends on the second derivative of the Fenchel conjugate of the generator function.

## 5 RESULTS

**Baselines** As baselines, we consider the CE, Forward loss (FL) Patrini et al. (2017), GCE Zhang & Sabuncu (2018), SCE Wang et al. (2019), NCE+RCE Ma et al. (2020), NCE+AGCE Zhou et al. (2021), ANL-CE/ANL-FL Ye et al. (2023), and RENT Bae et al. (2024). Additional baselines are considered in Appendix D.

**Implementation details** For the analysis on extreme noise rates, we use pre-trained feature extraction models downloaded from PyTorch Paszke et al. (2019), combined with logistic regression with $C = 10^{-4}$ and lbfgs solver. For evaluating the correction approaches, we use a ResNet34 He et al. (2016) for CIFAR-10 and a ResNet50 for CIFAR-100. For all the other experiments, unless differently specified, we use an 8-layer CNN as in Ye et al. (2023) for CIFAR-10 Krizhevsky et al. (2009) and CIFAR-10N Wei et al. (2021), and a ResNet34 for CIFAR-100 and CIFAR-100N. Optimization is executed using SGD with a momentum of 0.9. The learning rate is initially set to 0.02 and a cosine annealing scheduler Loshchilov & Hutter (2017) decays it during training. The tables report the mean over 5 independent runs of the code with different random seeds. Additional details are reported in Appendix D.1.

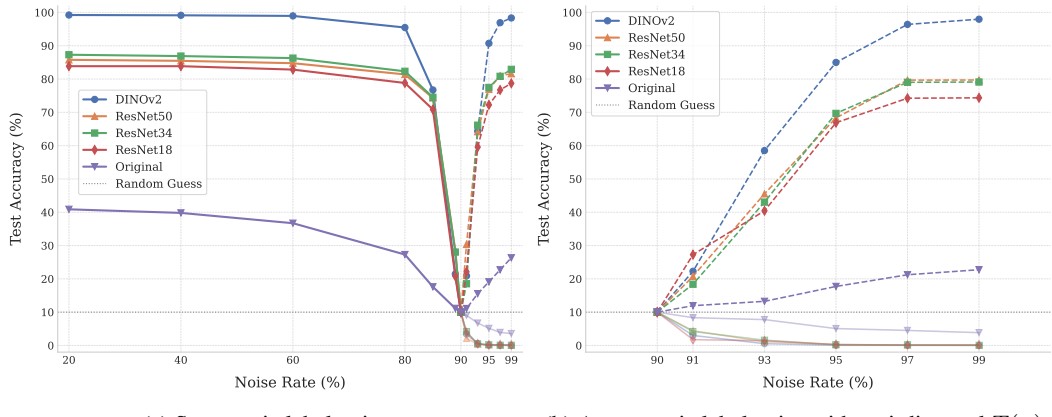

(a) Symmetric label noise         (b) Asymmetric label noise with anti-diagonal $T(\mathbf{x})$

Figure 2: A linear model trained on the features extracted by different methods, on CIFAR-10. *Solid* and **dashed** lines indicate that the class is predicted from the estimated noisy posterior using the standard *argmax* operation and the proposed **argmin** operation, respectively.

**Extremely high noise rates**   We demonstrate the numerical relevance of the novel theoretical results of Sec. 3 in Fig. 2. First, we test the validity of Corollary 3.2 in Fig. 2a, studying symmetric label noise on CIFAR-10. Then, we evaluate the more general case analyzed by Theorem 3.1 in Fig. 2b, where, for each row of the noise transition matrix, the elements of the main diagonal are smaller than all the other elements (but not symmetric noise). In both scenarios, we use different self-supervised learning-based feature extraction methods, i.e., DINOv2 Oquab et al. (2023), ResNet50, ResNet34, ResNet18 He et al. (2016), and then we perform classification on the extracted features using a noise ignorant logistic regression algorithm. In Fig. 2, the dashed lines indicate that the class is estimated by computing the argmin of the noisy posterior, and it is clear that this, contrarily to the standard argmax approach (identified by solid lines), leads to noise tolerant performance for extreme noise rates (i.e., $\eta > 0.9$).

In extreme label noise scenarios, the standard argmax-based class estimation approach leads to a test accuracy approaching $0\%$, whereas the proposed argmin-based approach yields a test accuracy approaching $100\%$. In addition, we also study the effectiveness of our theoretical results when using a neural network trained from scratch on a specific classification task. In fact, as explained at the end of Sec. 3, we cannot rely on foundation models for any task. We compare GAN-NPL and other APL-like losses for different noise rates in Fig. 3, which shows that GAN-NPL outperforms the other approaches for all the analyzed noise rates, especially for extremely high noise rates. For curiosity, we try to predict the class using the argmin method also for the other APL-like approaches, even though this is not proposed in the original papers and its correctness is not theoretically guaranteed, since the other APL-like losses do not target to

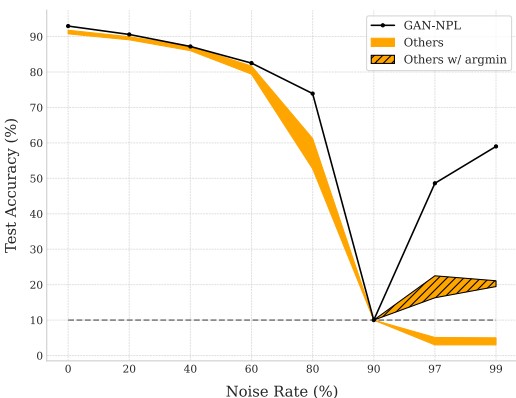

Figure 3: Comparison between GAN-NPL and other APL-like losses varying the noise rate, for CIFAR-10 with symmetric label noise.

estimate the posterior probability density. GAN-NPL outperforms other APL-like losses even when they predict the class using the argmin. Additional results on extreme noise rates are reported in Appendix D.

**Objective function and posterior correction**   The correction approaches rely on the knowledge of the transition matrix. We compare the performance of $f$-NPL and other correction methods using the true transition matrix and the transition matrix estimated through T estimation (referred to as Forward) Patrini et al. (2017) and DualT Yao et al. (2020). For multi-class classification,

Table 1: Test accuracy comparison between different correction approaches using various methods to estimate the transition matrix. $f$-NPL uses the GAN divergence.

| T Est. | Method | CIFAR-10 | | | | CIFAR-100 | | | |
|---|---|---|---|---|---|---|---|---|---|
| | | **SN** 20% | **SN** 50% | **AN** 20% | **AN** 40% | **SN** 20% | **SN** 50% | **AN** 20% | **AN** 40% |
| Forward | w/ FL | $87.50_{\pm 0.1}$ | $80.12_{\pm 0.4}$ | $87.61_{\pm 0.2}$ | $79.99_{\pm 0.4}$ | $60.20_{\pm 0.3}$ | $34.88_{\pm 0.8}$ | $60.07_{\pm 0.5}$ | $48.46_{\pm 0.4}$ |
| | w/ RENT | $86.89_{\pm 0.5}$ | $79.69_{\pm 0.5}$ | $86.90_{\pm 1.0}$ | $84.57_{\pm 1.9}$ | $57.81_{\pm 1.0}$ | $41.47_{\pm 1.9}$ | $58.43_{\pm 1.3}$ | $48.43_{\pm 1.8}$ |
| | **w/ $f$-NPL$_o$** | $92.14_{\pm 0.6}$ | $86.48_{\pm 0.6}$ | $92.20_{\pm 0.2}$ | $89.75_{\pm 0.7}$ | $72.44_{\pm 0.6}$ | $47.16_{\pm 0.7}$ | $73.30_{\pm 0.2}$ | $25.47_{\pm 1.2}$ |
| | **w/ $f$-NPL$_p$** | ***92.21***$_{\pm 0.1}$ | ***86.50***$_{\pm 0.5}$ | ***92.92***$_{\pm 0.2}$ | ***90.02***$_{\pm 0.8}$ | ***72.64***$_{\pm 0.2}$ | ***49.16***$_{\pm 0.3}$ | ***73.87***$_{\pm 0.5}$ | ***57.18***$_{\pm 1.1}$ |
| DualT | w/ FL | $88.08_{\pm 0.3}$ | $81.65_{\pm 0.5}$ | $88.43_{\pm 0.4}$ | $70.86_{\pm 0.7}$ | $61.74_{\pm 0.5}$ | $38.14_{\pm 0.5}$ | $60.30_{\pm 0.2}$ | $48.17_{\pm 0.6}$ |
| | w/ RENT | $87.74_{\pm 0.6}$ | $80.50_{\pm 0.8}$ | $86.43_{\pm 0.9}$ | $81.18_{\pm 2.0}$ | $58.81_{\pm 0.9}$ | $42.46_{\pm 1.7}$ | $58.58_{\pm 0.8}$ | $51.48_{\pm 1.7}$ |
| | **w/ $f$-NPL$_o$** | $91.88_{\pm 0.4}$ | $83.17_{\pm 1.0}$ | $91.51_{\pm 0.3}$ | $84.11_{\pm 0.6}$ | $74.10_{\pm 0.4}$ | $49.06_{\pm 0.5}$ | $72.08_{\pm 0.5}$ | $54.61_{\pm 0.9}$ |
| | **w/ $f$-NPL$_p$** | ***92.41***$_{\pm 0.3}$ | ***85.15***$_{\pm 0.2}$ | ***93.12***$_{\pm 0.2}$ | ***88.55***$_{\pm 0.9}$ | ***74.52***$_{\pm 0.6}$ | ***57.18***$_{\pm 0.4}$ | ***74.39***$_{\pm 0.6}$ | ***64.85***$_{\pm 0.9}$ |
| True $T$ | w/FL | $87.61_{\pm 0.3}$ | $81.76_{\pm 0.6}$ | $87.66_{\pm 0.3}$ | $82.42_{\pm 0.6}$ | $62.59_{\pm 0.7}$ | $52.08_{\pm 0.6}$ | $62.46_{\pm 0.5}$ | $57.83_{\pm 0.8}$ |
| | w/RENT | $86.57_{\pm 0.5}$ | $79.96_{\pm 0.9}$ | $86.22_{\pm 1.2}$ | $79.12_{\pm 2.1}$ | $61.07_{\pm 1.4}$ | $52.18_{\pm 2.0}$ | $60.81_{\pm 0.9}$ | $56.77_{\pm 2.1}$ |
| | **w/ $f$-NPL$_o$** | $91.96_{\pm 0.7}$ | $84.03_{\pm 1.2}$ | ***92.70***$_{\pm 0.4}$ | ***88.30***$_{\pm 0.6}$ | $73.26_{\pm 0.9}$ | $51.10_{\pm 0.9}$ | $73.44_{\pm 0.3}$ | ***57.97***$_{\pm 0.7}$ |
| | **w/ $f$-NPL$_p$** | ***92.14***$_{\pm 0.3}$ | ***84.56***$_{\pm 0.7}$ | $92.51_{\pm 0.1}$ | $87.92_{\pm 1.1}$ | ***73.31***$_{\pm 0.2}$ | ***52.20***$_{\pm 0.8}$ | ***73.84***$_{\pm 0.3}$ | $57.43_{\pm 0.8}$ |
| No Corr. | $f$-**NPL** | **92.59**$_{\pm 0.2}$ | **86.54**$_{\pm 0.4}$ | $93.07_{\pm 0.3}$ | $88.01_{\pm 0.9}$ | **73.05**$_{\pm 0.3}$ | **52.42**$_{\pm 0.4}$ | $73.16_{\pm 0.4}$ | $56.59_{\pm 0.7}$ |

the correction approaches are tested on CIFAR-10 and CIFAR-100 for symmetric and asymmetric uniform off-diagonal label noise in Tab. 1, where ***bold-italic*** represents the best accuracy for a specific label noise setting and transition matrix estimation method and **bold** represents the best accuracy for a specific label noise over all $T$ estimation methods. Tab. 1 shows that $f$-NPL's correction approaches outperform other methods that rely on objective function correction. We noticed that, on average, $f$-NPL$_p$ achieves slightly higher accuracy than $f$-NPL$_o$. This, coupled with the lower computational complexity of $f$-NPL$_p$ compared to $f$-NPL$_o$ (as detailed in Appendix D.2.4), suggests that it is the superior correction approach. Additional results are reported in Appendix D, including the evaluation of $f$-NPL on the binary classification scenario and for sparse label noise, demonstrating the effectiveness of the correction approaches in these settings.

Table 2: Test accuracy on ILSVRC12 and Mini WebVision.

| Dataset | CE | GCE | SCE | NCE+RCE | NCE+AGCE | ANL-CE | ANL-FL | **SL-NPL** | **GAN-NPL** |
|---|---|---|---|---|---|---|---|---|---|
| ILSVRC12 | 58.64 | 56.56 | 62.60 | 62.40 | 60.76 | 65.00 | 65.56 | 74.53 | **74.56** |
| WebVision | 61.20 | 59.44 | 68.00 | 64.92 | 63.92 | 67.44 | 68.32 | 77.27 | **79.53** |

**Comparison between $f$-NPL and other APL-like losses** Since $f$-NPL possesses APL-like properties, we perform a comparative analysis with the existing APL-like losses. We train a ResNet50 on mini WebVision Li et al. (2017) and then test the trained network on the validation datasets of mini WebVision and ImageNet ILSVRC12 Krizhevsky et al. (2012) (Tab. 2). $f$-NPL significantly outperforms the other APL-like methods. For space limitations, we report in Appendix D experiments on various additional scenarios, which demonstrate that $f$-NPL performs better than the other APL-like objective functions in the presence of symmetric, asymmetric, and real-world label noise. The efficacy of $f$-NPL compared to the other APL-like methods can be attributed to the fact that other APL-like methods rely on the sum of two independent active and passive losses, while the $f$-NPL framework implicitly defines a relationship between active and passive terms, leading to the estimation of the posterior.

## 6 CONCLUSIONS

In this paper, we prove that deep learning models estimating the posterior can accurately predict the true class under extremely high label noise rates, even when not performing correction methods and, in certain cases, without knowledge of the noise transition matrix. Building on this finding, we propose an $f$-divergence-based noisy posterior learning ($f$-NPL) technique, which is also intrinsically robust to label noise under specific assumptions. To address scenarios where these assumptions are violated, we present two correction methods: an objective function correction approach and a novel posterior estimator correction technique. Finally, through the experimental results, we demonstrate the correctness of the theoretical claims on extreme noise rates, and the effectiveness of $f$-NPL for synthetic and real-world label noise.

REPRODUCIBILITY STATEMENT

For theoretical results, we provide the extensive proofs in Appendix B. We describe the details to reproduce the numerical results provided in the paper at the beginning of Sec. 5 and in Appendix D.1. Furthermore, we provide the code as supplementary material.

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

Table 3: $f$-divergences table. The corresponding $f$-divergences are: Kullback-Leibler, GAN, Shifted Log.

| Name | $f(u)$ | $f^*(t)$ | $D^{\diamond}(p_{Y|X})$ |
|------|--------|----------|------------------------|
| KL | $u\log(u)$ | $\exp(t-1)$ | $\log(p_{Y|X})+1$ |
| GAN | $u\log(u)-(u+1)\log(u+1)$ | $-\log(1-\exp(t))$ | $\log(p_{Y|X}/(p_{Y|X}+1))$ |
| SL | $-\log(u+1)$ | $-(\log(-t)+t)$ | $-1/(p_{Y|X}+1)$ |

# A ADDITIONAL DETAILS ON THE OBJECTIVE FUNCTIONS

The generator functions of the $f$-divergences used in this paper are reported in Tab. 3, along with their Fenchel conjugate functions $f^*$, and the optimal value achieved by the neural network at convergence $D^{\diamond} = f'(p_{XY}/p_X)$.

In the following, we list the objective functions of $f$-NPL corresponding to different $f$-divergences. For the experiments on the objective function and posterior correction approaches, following the work in Nowozin et al. (2016), the neural network's output is expressed as $D = g_f(v)$, where $v$ is a linear layer output of the neural network, and $g_f(\cdot)$ is a monotonically increasing function as defined in Nowozin et al. (2016). However, we noticed that for datasets with a large amount of classes, like CIFAR-100, the training sometimes fails when using these objective functions. In those cases, we apply a change of variable $D = r(D')$ that improves the training process, where $D$ is not expressed based on $g_f(\cdot)$. For all the objective functions, we use the following notation: $\mathbf{1}_K(y_{\mathbf{x}})$ is a one-hot column vector equal to 1 in correspondence of the label $y_{\mathbf{x}}$, $\mathbf{1}_K$ is a column vector of 1s of length $K$.

**Kullback-Leibler divergence** The objective function corresponding to the KL divergence is

$$\mathcal{J}_{KL}(D) = \mathbb{E}_{XY}[D(\mathbf{x})\mathbf{1}_K(y_{\mathbf{x}})] + \mathbb{E}_X\left[\sum_{i=1}^K -e^{D(\mathbf{x},i)-1}\right]. \tag{17}$$

Substituting $D^{\diamond}$ from Tab. 3, we get

$$\mathcal{J}_{KL}(D^{\diamond}) = \mathbb{E}_{XY}\left[\log\left(p_{Y|X}(y_{\mathbf{x}}|\mathbf{x})\right)\right] + \mathbb{E}_X\left[\sum_{i=1}^K\left(-p_{Y|X}(i|\mathbf{x})\right)\right]. \tag{18}$$

Using the change of variable $D = \log(D')+1$ (thus $D'(\mathbf{x}) = [p_{Y|X}(1|\mathbf{x}),\ldots,p_{Y|X}(K|\mathbf{x})]$), the objective function rewrites as

$$\mathcal{J}_{KL}(D') = \mathbb{E}_{XY}[\log(D'(\mathbf{x}))\mathbf{1}_K(y_{\mathbf{x}})] + \mathbb{E}_X[-D'(\mathbf{x})\mathbf{1}_K]. \tag{19}$$

**GAN divergence** The objective function corresponding to the GAN divergence is

$$\mathcal{J}_{GAN}(D) = \mathbb{E}_{XY}[D(\mathbf{x})\mathbf{1}_K(y_{\mathbf{x}})] + \mathbb{E}_X\left[\sum_{i=1}^K\log\left(1-e^{D(\mathbf{x},i)}\right)\right], \tag{20}$$

Substituting $D^{\diamond}$ from Tab. 3, we get

$$\mathcal{J}_{GAN}(D^{\diamond}) = \mathbb{E}_{XY}\left[\log\left(\frac{p_{Y|X}(y_{\mathbf{x}}|\mathbf{x})}{p_{Y|X}(y_{\mathbf{x}}|\mathbf{x})+1}\right)\right] + \mathbb{E}_X\left[\sum_{i=1}^K\log\left(\frac{1}{p_{Y|X}(i|\mathbf{x})+1}\right)\right]. \tag{21}$$

Using the change of variable $D = \log(D'/(D'+1))$, the objective function writes as

$$\mathcal{J}_{GAN}(D') = \mathbb{E}_{XY}\left[\log\left(\frac{D'(\mathbf{x})}{D'(\mathbf{x})+1}\right)\mathbf{1}_K(y_{\mathbf{x}})\right] + \mathbb{E}_X\left[\sum_{i=1}^K\log\left(\frac{1}{D'(\mathbf{x},i)+1}\right)\right]. \tag{22}$$

**Shifted log divergence** The objective function corresponding to the SL divergence is

$$\mathcal{J}_{SL}(D) = \mathbb{E}_{XY}\left[D(\mathbf{x})\mathbf{1}_K(y_{\mathbf{x}})\right] + \mathbb{E}_X\left[-\sum_{i=1}^K \left(-(\log(-D(\mathbf{x},i)) + D(\mathbf{x},i))\right)\right]. \tag{23}$$

Substituting $D^\diamond$ from Tab. 3, we get

$$\mathcal{J}_{SL}(D^\diamond) = \mathbb{E}_{XY}\left[-\frac{1}{p_{Y|X}(y_{\mathbf{x}}|\mathbf{x})+1}\right]$$

$$+ \mathbb{E}_X\left[\sum_{i=1}^K \left(-\frac{1}{p_{Y|X}(i|\mathbf{x})+1} + \log\left(\frac{1}{p_{Y|X}(i|\mathbf{x})+1}\right)\right)\right]. \tag{24}$$

Using the change of variable $D = -1/(D'+1)$, the objective function writes as

$$\mathcal{J}_{SL}(D') = \mathbb{E}_{XY}\left[-\frac{1}{D'(\mathbf{x})+1}\mathbf{1}_K(y_{\mathbf{x}})\right] + \mathbb{E}_X\left[\sum_{i=1}^K \left(-\frac{1}{D'(\mathbf{x},i)+1} + \log\left(\frac{1}{D'(\mathbf{x},i)+1}\right)\right)\right]. \tag{25}$$

### A.1 CONNECTION WITH EMPIRICAL RISK MINIMIZATION (ERM)

$f$-NPL can be framed in the ERM framework as a specific class of losses depending on $f$. The ERM can be written as

$$\mathcal{J}(h) = \frac{1}{N}\sum_{i=1}^N l(h(\mathbf{x}_i), y_i), \tag{26}$$

where $l(\cdot)$ is the loss, $h(\cdot)$ is the network's output, $\mathbf{x}_i$ and $y_i$ are the input and label of sample $i$, respectively, and $N$ is the number of samples. To express $f$-NPL in terms of the ERM framework we rewrite equation 6 using

$$l(h(\mathbf{x}_i), y_i) = -\left(h(\mathbf{x}_i)_{y_{\mathbf{x}_i}} - \sum_{j=1}^K f^*(h(\mathbf{x}_i)_j)\right), \tag{27}$$

where $h(\mathbf{x})_j$ refers to the $j$-th output neuron of $h(\mathbf{x})$. From this expression, we notice the connection between ERM and $f$-NPL, and we see that the class of losses imposed by $f$-NPL depends on $f^*$. A typical loss belonging to the ERM framework used for classification is the CE, for which the network's output coincides with the estimated posterior. $f$-NPL leverages Nguyen et al. (2010) to return the posterior as a function of the network's output as in equation 7. In particular, $f$-NPL leverages the theoretical results in Novello & Tonello (2024), which relies on Nguyen et al. (2010) as follows: equation 7 expresses the posterior as the density-ratio learned by the neural network trained with equation 6 (i.e., $p_{Y|X} = p_{XY}/p_X$), and uses such an estimated density-ratio to perform classification.

## B PROOFS

### B.1 PROOF OF THEOREM 3.1

**Theorem 3.1.** *Let $p_{Y|X}(\cdot|\mathbf{x}) \in \{\mathbf{e}_1, \ldots, \mathbf{e}_K\}$ be a one-hot vector, and assume the diagonal elements of $T(\mathbf{x})$ minimize their rows, then*

$$\arg\max_y p_{Y|X}(y|\mathbf{x}) = \arg\min_y p_{Y_\eta|X}(y|\mathbf{x}). \tag{28}$$

*Proof.* If the posterior is a one-hot vector, let $p_{Y|X}(y|\mathbf{x}) = \mathbf{e}_y$ for some $y \in \{1, \ldots, K\}$. The noisy posterior is obtained as $p_{Y_\eta|X}(\cdot|\mathbf{x}) = T(\mathbf{x})^T p_{Y|X}(\cdot|\mathbf{x}) = [T(\mathbf{x})]_{y,:}^T$, where $[T(\mathbf{x})]_{y,:}^T$ indicates the entire $y$-th row of $T(\mathbf{x})$, transposed. Then,

$$\arg\min_a p_{Y_\eta|X}(a|\mathbf{x}) = \arg\min_a [T(\mathbf{x})]_{y,:}^T = y = \arg\max_a p_{Y|X}(a|\mathbf{x}), \tag{29}$$

which derives from the Theorem's assumption about the fact that the diagonal elements of the transition matrix minimize their rows, and from the fact that, during the proof, we set $p_{Y|X}(y|\mathbf{x}) = \mathbf{e}_y$. □

## B.2 PROOF OF COROLLARY 3.2

**Corollary 3.2.** *For symmetric label noise, when $\eta > \frac{K-1}{K}$, the class minimizing the noisy posterior coincides with the class predicted by the optimal Bayes classifier in the absence of label noise.*

*Proof.* First, we notice that if the diagonal elements of $T(\mathbf{x})$ minimize their rows, then the noise rate exceeds the threshold $\frac{K-1}{K}$. In fact, the value of the elements in the main diagonal is $1 - \eta$, and all the elements not in the main diagonal coincide with $\frac{\eta}{K-1}$. If $\eta > \frac{K-1}{K}$, each element outside the main diagonal is greater than $1/K$, implying that the elements along the main diagonal are smaller than $1/K$. Thus, for symmetric label noise, when $\eta > \frac{K-1}{K}$, the elements on the main diagonal minimize their rows.

Below we provide a deeper analysis that analysis also the case in which the true posterior is not a one-hot vector. When $\eta = \frac{K-1}{K}$,

$$p_{Y_\eta|X}(i|\mathbf{x}) = \frac{1}{K} \quad \forall i \in [K], \tag{30}$$

which implies that it is not possible to find the class maximizing $p_{Y|X}(\cdot|\mathbf{x})$ given $p_{Y_\eta|X}(\cdot|\mathbf{x})$.

When $\frac{K-1}{K} < \eta < 1$,

$$p_{Y_\eta|X}(i|\mathbf{x}) = -\alpha p_{Y|X}(i|\mathbf{x}) + \beta \quad \forall i \in [K], \tag{31}$$

with $\alpha > 0, \beta > 0$. Trivially, $\arg\max_i p_{Y_\eta|X}(i|\mathbf{x}) \neq \arg\max_i p_{Y|X}(i|\mathbf{x})$. Let us first consider the single-label case for an easier understanding; then we will extend this to the general case.
For the single-label case, without loss of generality, assume $p_{Y|X} = [0, \ldots, 0, \underbrace{1}_{i_M\text{-th pos.}}, 0, \ldots, 0]$,
$i_M \in [K]$. Then, $p_{Y_\eta|X}(i_M|\mathbf{x}) = -\alpha + \beta$, while $p_{Y_\eta|X}(i|\mathbf{x}) = \beta, \forall i \in [K], i \neq i_M$. Since $\alpha > 0$, $p_{Y_\eta|X}(i_M|\mathbf{x}) < p_{Y_\eta|X}(i|\mathbf{x}), \forall i \in [K], i \neq i_M$.
For the general case, $p_{Y|X} = [p_1, p_2, \ldots, p_K]$. Let us assume that $i_M = \arg\max_i p_{Y|X}(i|\mathbf{x})$. Therefore, $p_{Y|X}(i_M|\mathbf{x}) > p_{Y|X}(j|\mathbf{x}), \forall j \in [K], j \neq i_M$. Thus, $\beta - \alpha p_{Y|X}(i_M|\mathbf{x}) < \beta - \alpha p_{Y|X}(j|\mathbf{x}), \forall j \in [K], j \neq i_M$, which coincides with $p_{Y_\eta|X}(i_M|\mathbf{x}) < p_{Y_\eta|X}(j|\mathbf{x}), \forall j \in [K]$, $j \neq i_M$, implying that $i_M = \arg\min_i p_{Y_\eta|X}(i|\mathbf{x})$.
Finally, for symmetric label noise with $\frac{K-1}{K} < \eta < 1$,

$$\hat{y}_\mathbf{x} = \arg\max_{y_\mathbf{x} \in \mathcal{A}_y} p_{Y|X}(y_\mathbf{x}|\mathbf{x}) = \arg\min_{y_\mathbf{x} \in \mathcal{A}_y} p_{Y_\eta|X}(y_\mathbf{x}|\mathbf{x}) = \hat{y}_\mathbf{x}^\eta. \tag{32}$$

In summary, if $\eta < \frac{K-1}{K}$, which is the case studied in most of the literature, estimating the noisy posterior leads to an algorithm robust to symmetric label noise. Otherwise, when $\frac{K-1}{K} < \eta < 1$, it is possible to estimate the true label by finding the class that minimizes the estimated posterior (i.e., by finding the argmin of $p_{Y_\eta|X}(y_\mathbf{x}|\mathbf{x})$). This procedure is significantly different from the posterior correction approach: for the posterior correction, it is crucial to know the exact value of the noise transition probabilities, while in this case it is sufficient to know that the noise rate is larger than $\frac{K-1}{K}$. □

## B.3 PROOF OF THEOREM 4.1

**Theorem 4.1.** *For binary classification, the relationship between the value of the objective function in the presence ($\mathcal{J}_f^\eta(D)$) and absence ($\mathcal{J}_f(D)$) of label noise, given the same parametric function $T$, is*

$$\mathcal{J}_f^\eta(D) = (1 - e_0 - e_1)\mathcal{J}_f(D) + B_f(D), \tag{33}$$

*where*

$$B_f(D) \triangleq \mathbb{E}_X\left[e_0 D(\mathbf{x}, 0) + e_1 D(\mathbf{x}, 1) - (e_0 + e_1)\sum_{i=0}^{1} f^*(D(\mathbf{x}, i))\right] \tag{34}$$

*is a bias term.*

*Proof.* The value of the objective function in the presence of label noise, according to equation 6, is obtained as

$$\mathcal{J}_f^\eta(D) = \mathbb{E}_{XY_\eta}\Big[D(\mathbf{x}, \tilde{y}_\mathbf{x})\Big] - \mathbb{E}_X\Big[\sum_{i=0}^{1} f^*(D(\mathbf{x}, i))\Big]. \tag{35}$$

Given that the label noise is conditionally independent on $X$, the first term in equation 35 rewrites as

$$\mathbb{E}_{XY_\eta}[D(\mathbf{x}, \tilde{y}_\mathbf{x})] = \mathbb{E}_Y\mathbb{E}_{X|Y}\mathbb{E}_{Y_\eta|Y}\Big[D(\mathbf{x}, \tilde{y}_\mathbf{x})\Big] \tag{36}$$

$$= p_Y(0)\mathbb{E}_{X|Y=0}\left[\mathbb{P}[Y_\eta = 0|Y = 0]D(\mathbf{x}, 0) + \mathbb{P}[Y_\eta = 1|Y = 0]D(\mathbf{x}, 1)\right]$$
$$+ (1 - p_Y(0))\mathbb{E}_{X|Y=1}\left[\mathbb{P}[Y_\eta = 0|Y = 1]D(\mathbf{x}, 0) + \mathbb{P}[Y_\eta = 1|Y = 1]D(\mathbf{x}, 1)\right] \tag{37}$$

$$= p_Y(0)\mathbb{E}_{X|Y=0}\left[(1 - e_1)D(\mathbf{x}, 0) + e_1 D(\mathbf{x}, 1)\right]$$
$$+ (1 - p_Y(0))\mathbb{E}_{X|Y=1}\left[e_0 D(\mathbf{x}, 0) + (1 - e_0)D(\mathbf{x}, 1)\right] \tag{38}$$

$$= p_Y(0)\mathbb{E}_{X|Y=0}\left[(1 - e_0 - e_1)D(\mathbf{x}, 0) + e_0 D(\mathbf{x}, 0) + e_1 D(\mathbf{x}, 1)\right]$$
$$+ (1 - p_Y(0))\mathbb{E}_{X|Y=1}\left[e_0 D(\mathbf{x}, 0) + (1 - e_1 - e_0)D(\mathbf{x}, 1) + e_1 D(\mathbf{x}, 1)\right] \tag{39}$$

$$= p_Y(0)\mathbb{E}_{X|Y=0}\left[(1 - e_0 - e_1)D(\mathbf{x}, 0)\right]$$
$$+ (1 - p_Y(0))\mathbb{E}_{X|Y=1}\left[(1 - e_0 - e_1)D(\mathbf{x}, 1)\right]$$
$$+ p_Y(0)\mathbb{E}_{X|Y=0}\left[e_0 D(\mathbf{x}, 0) + e_1 D(\mathbf{x}, 1)\right]$$
$$+ (1 - p_Y(0))\mathbb{E}_{X|Y=1}\left[e_0 D(\mathbf{x}, 0) + e_1 D(\mathbf{x}, 1)\right] \tag{40}$$

$$= (1 - e_0 - e_1)\mathbb{E}_{XY}[D(\mathbf{x}, y_\mathbf{x})] + \mathbb{E}_X\left[e_0 D(\mathbf{x}, 0) + e_1 D(\mathbf{x}, 1)\right] \tag{41}$$

and

$$\mathbb{E}_X\left[f^*(D(\mathbf{x}, 0)) + f^*(D(\mathbf{x}, 1))\right] = (1 - e_0 - e_1)\mathbb{E}_X\left[f^*(D(\mathbf{x}, 0)) + f^*(D(\mathbf{x}, 1))\right]$$
$$+ (e_0 + e_1)\mathbb{E}_X\left[f^*(D(\mathbf{x}, 0)) + f^*(D(\mathbf{x}, 1))\right]. \tag{42}$$

The second term is not affected by the presence of label noise.

Subtracting the first RHS term in equation 42 to the first RHS term in equation 41, we get

$$(1 - e_0 - e_1)\mathbb{E}_{XY}[D(\mathbf{x}, y_\mathbf{x})] - (1 - e_0 - e_1)\mathbb{E}_X\left[\sum_{i=0}^{1} f^*(D(\mathbf{x}, i))\right] = (1 - e_0 - e_1)\mathcal{J}_f(D), \tag{43}$$

where $\mathcal{J}_f(D)$ is the value of the objective function when the training is done in the absence of label noise. Subtracting the second RHS term in equation 42 to the second RHS term in equation 41, we get

$$\mathbb{E}_X\left[e_0 D(\mathbf{x}, 0) + e_1 D(\mathbf{x}, 1) - (e_0 + e_1)\sum_{i=0}^{1} f^*(D(\mathbf{x}, i))\right] \triangleq B_f(D) \tag{44}$$

Putting all together, we obtain the theorem's claim. $\qquad\square$

### B.4    PROOF OF THEOREM 4.3

**Theorem 4.3.** *For multi-class asymmetric uniform off-diagonal label noise, the relationship between the value of the objective function in the presence ($\mathcal{J}_f^\eta(D)$) and absence ($\mathcal{J}_f(D)$) of label noise, given the same parametric function $D$, is*

$$\mathcal{J}_f^\eta(D) = \left(1 - \sum_{j=1}^{K} e_j\right)\mathcal{J}_f(D) + B_f(D), \tag{45}$$

*where*

$$B_f(D) \triangleq \mathbb{E}_X\left[\sum_{j=1}^{K}\left(e_j D(\mathbf{x}, j) - \left(\sum_{i=1}^{K} e_i\right)f^*(D(\mathbf{x}, j))\right)\right]. \tag{46}$$

*Proof.* Let $p_i \triangleq P(Y = i)$. We have $\tilde{p}_i \triangleq P(\tilde{Y} = i) = \left(1 - \sum_{j \neq i} e_j\right) p_i + e_i \sum_{j \neq i} p_j$. The objective function in the presence of label noise is

$$\mathcal{J}_f^\eta(D) = \mathbb{E}_{XY_\eta}\left[D(\mathbf{x}, \tilde{y}_\mathbf{x})\right] - \mathbb{E}_X\left[\sum_{i=1}^K f^*(D(\mathbf{x}, i))\right]. \tag{47}$$

The first term can be rewritten as

$$\mathbb{E}_{XY_\eta}\left[D(\mathbf{x}, \tilde{y}_\mathbf{x})\right] = \mathbb{E}_Y \mathbb{E}_{X|Y} \mathbb{E}_{Y_\eta|Y}\left[D(\mathbf{x}, \tilde{y}_\mathbf{x})\right] \tag{48}$$

$$= \sum_{i=1}^K p_i \mathbb{E}_{X|Y=i}\left[\left(1 - \sum_{j \neq i} e_j\right) D(\mathbf{x}, i) + \sum_{j \neq i} e_j D(\mathbf{x}, j)\right] \tag{49}$$

$$= \sum_{i=1}^K p_i \mathbb{E}_{X|Y=i}\left[\left(1 - \sum_{j=1}^K e_j\right) D(\mathbf{x}, i) + \sum_{j=1}^K e_j D(\mathbf{x}, j)\right] \tag{50}$$

$$= \left(1 - \sum_{j=1}^K e_j\right) \mathbb{E}_{XY}\left[D(\mathbf{x}, y_\mathbf{x})\right] + \sum_{j=1}^K e_j \mathbb{E}_X\left[D(\mathbf{x}, j)\right]. \tag{51}$$

As in the binary case, the second term of equation 47 is not influenced by the presence of label noise. Merging the two terms we obtain the theorem's claim

$$\mathcal{J}_f^\eta(D) = \left(1 - \sum_{j=1}^K e_j\right) \mathbb{E}_{XY}\left[D(\mathbf{x}, y_\mathbf{x})\right] + \sum_{j=1}^K e_j \mathbb{E}_X\left[D(\mathbf{x}, j)\right] - \mathbb{E}_X\left[\sum_{j=1}^K f^*(D(\mathbf{x}, j))\right] \tag{52}$$

$$= \left(1 - \sum_{j=1}^K e_j\right) \mathbb{E}_{XY}\left[D(\mathbf{x}, y_\mathbf{x})\right] - \left(1 - \sum_{j=1}^K e_j\right) \mathbb{E}_X\left[\sum_{j=1}^K f^*(D(\mathbf{x}, j))\right]$$

$$+ \underbrace{\sum_{j=1}^K \left(e_j \mathbb{E}_X\left[D(\mathbf{x}, j)\right]\right) - \left(\sum_{j=1}^K e_j\right) \mathbb{E}_X\left[\sum_{j=1}^K f^*(D(\mathbf{x}, j))\right]}_{\triangleq B_f(D)} \tag{53}$$

$$= \left(1 - \sum_{j=1}^K e_j\right) \mathcal{J}_f(D) + B_f(D). \tag{54}$$

$\square$

### B.5 PROOF OF THEOREM 4.4

**Theorem 4.4.** *For the binary classification case, the posterior estimator in the presence of label noise is related to the clean posterior estimator as*

$$\hat{p}_{Y|X}^\eta(i|\mathbf{x}) = (f^*)'(D_\eta^\diamond(\mathbf{x}, i))$$

$$= (1 - e_0 - e_1)\hat{p}_{Y|X}(i|\mathbf{x}) + e_i, \tag{55}$$

$\forall i \in \{0, 1\}$.

*Proof.* The expression of $\mathcal{J}_f(D)$ can be rewritten as

$$\mathcal{J}_f(D) = \mathbb{E}_{XY}\left[D(\mathbf{x}, y_\mathbf{x})\right] - \mathbb{E}_X\left[\sum_{i=0}^{1} f^*(D(\mathbf{x}, i))\right] \tag{56}$$

$$= \mathbb{E}_{XY}\left[D(\mathbf{x}, y_\mathbf{x})\right] - \mathbb{E}_X\left[f^*(D(\mathbf{x}, 0)) + f^*(D(\mathbf{x}, 1))\right] \tag{57}$$

$$= \mathbb{E}_Y\left[\mathbb{E}_{X|Y}\left[D(\mathbf{x}, y_\mathbf{x})\right]\right] - \mathbb{E}_X\left[f^*(D(\mathbf{x}, 0)) + f^*(D(\mathbf{x}, 1))\right] \tag{58}$$

$$= p_Y(0)\left[\mathbb{E}_{X|Y=0}\left[D(\mathbf{x}, 0)\right]\right] + p_Y(1)\left[\mathbb{E}_{X|Y=1}\left[D(\mathbf{x}, 1)\right]\right]$$
$$- \mathbb{E}_X\left[f^*(D(\mathbf{x}, 0)) + f^*(D(\mathbf{x}, 1))\right] \tag{59}$$

$$= \underbrace{p_Y(0)\left[\mathbb{E}_{X|Y=0}\left[D(\mathbf{x}, 0)\right]\right] - \mathbb{E}_X\left[f^*(D(\mathbf{x}, 0))\right]}_{\triangleq \mathcal{J}_{f,0}(D)}$$

$$+ \underbrace{p_Y(1)\left[\mathbb{E}_{X|Y=1}\left[D(\mathbf{x}, 1)\right]\right] - \mathbb{E}_X\left[f^*(D(\mathbf{x}, 1))\right]}_{\triangleq \mathcal{J}_{f,1}(D)}. \tag{60}$$

Similarly, the bias term can be rewritten as

$$B_f(D) = \mathbb{E}_X\left[e_0 D(\mathbf{x}, 0) + e_1 D(\mathbf{x}, 1) - (e_0 + e_1)\big(f^*(D(\mathbf{x}, 0)) + f^*(D(\mathbf{x}, 1))\big)\right] \tag{61}$$

$$= \underbrace{\mathbb{E}_X\left[e_0 D(\mathbf{x}, 0) - (e_0 + e_1)f^*(D(\mathbf{x}, 0))\right]}_{\triangleq B_{f,0}(D)} + \underbrace{\mathbb{E}_X\left[e_1 D(\mathbf{x}, 1) - (e_0 + e_1)f^*(D(\mathbf{x}, 1))\right]}_{\triangleq B_{f,1}(T)}. \tag{62}$$

Merging the two expressions for $\mathcal{J}_f$ and $B_f$ with Theorem 4.1, the objective function in presence of label noise becomes

$$\mathcal{J}_f^\eta(D) = (1 - e_0 - e_1)\mathcal{J}_f(D) + B_f(D) \tag{63}$$

$$= (1 - e_0 - e_1)(\mathcal{J}_{f,0}(D) + \mathcal{J}_{f,1}(D)) + B_{f,0}(D) + B_{f,1}(D) \tag{64}$$

$$= \underbrace{(1 - e_0 - e_1)\mathcal{J}_{f,0}(D) + B_{f,0}(D)}_{\triangleq \mathcal{J}_{f,0}^\eta(D)} + \underbrace{(1 - e_0 - e_1)\mathcal{J}_{f,1}(D) + B_{f,1}(D)}_{\triangleq \mathcal{J}_{f,1}^\eta(D)}. \tag{65}$$

$B_{f,0}(D)$ and $(1 - e_0 - e_1)\mathcal{J}_{f,0}(D)$ are concave in $D$. Therefore, $\mathcal{J}_{f,0}^\eta(D)$ is concave in $D$ because sum of concave functions. Since $\mathcal{J}_{f,0}^\eta(D)$ is concave, the optimal convergence condition of $D$ is achieved imposing the first derivative of $\mathcal{J}_{f,0}^\eta(D)$ equal to 0. $\mathcal{J}_{f,0}^\eta(D)$ can be rewritten as

$$\mathcal{J}_{f,0}^\eta(D) = (1 - e_0 - e_1)(p_Y(0)\left[\mathbb{E}_{X|Y=0}\left[D(\mathbf{x}, 0)\right]\right] - \mathbb{E}_X\left[f^*(D(\mathbf{x}, 0))\right]) \tag{66}$$

$$+ \mathbb{E}_X\left[e_0 D(\mathbf{x}, 0) - (e_0 + e_1)f^*(D(\mathbf{x}, 0))\right] \tag{67}$$

$$= \int_{\mathcal{X}} (1 - e_0 - e_1)(p_Y(0)p_{X|Y}(\mathbf{x}|0)D(\mathbf{x}, 0) - p_X(\mathbf{x})f^*(D(\mathbf{x}, 0)))$$
$$+ p_X(\mathbf{x})e_0 D(\mathbf{x}, 0) - p_X(\mathbf{x})(e_0 + e_1)f^*(D(\mathbf{x}, 0))d\mathbf{x}. \tag{68}$$

Thus, imposing the first derivative w.r.t. $D$ equals to 0 yields

$$(f^*)'(D(\mathbf{x}, 0)) = (1 - e_0 - e_1)p_{Y|X}(0|\mathbf{x}) + e_0. \tag{69}$$

Since $(f^*)'(t) = (f')^{-1}(t)$,

$$D_\eta^\diamond(\mathbf{x}, 0) = f'((1 - e_0 - e_1)p_{Y|X}(0|\mathbf{x}) + e_0), \tag{70}$$

where $D_\eta^\diamond(\mathbf{x}, 0)$ indicates the neural network at convergence. Therefore, the posterior estimator obtained in the presence of label noise reads as

$$\hat{p}_{Y|X}^\eta(0|\mathbf{x}) = (f^*)'(D_\eta^\diamond(\mathbf{x}, 0)) = (1 - e_0 - e_1)p_{Y|X}(0|\mathbf{x}) + e_0. \tag{71}$$

The same calculations can be done for $\mathcal{J}_{f,1}^{\eta}(D)$, leading to

$$\hat{p}_{Y|X}^{\eta}(1|\mathbf{x}) = (f^*)'(D_{\eta}^{\diamond}(\mathbf{x},1)) = (1 - e_0 - e_1)p_{Y|X}(1|\mathbf{x}) + e_1. \tag{72}$$

$\square$

### B.6  PROOF OF THEOREM 4.6

**Theorem 4.6.** *For multi-class asymmetric uniform off-diagonal label noise, the relationship between the posterior estimator in the presence and absence of label noise is*

$$\hat{p}_{Y|X}^{\eta}(i|\mathbf{x}) = (f^*)'(D_{\eta}^{\diamond}(\mathbf{x},i))$$

$$= \left(1 - \sum_{j=1}^{K} e_j\right)\hat{p}_{Y|X}(i|\mathbf{x}) + e_i, \tag{73}$$

$\forall i \in \{1,\ldots,K\}$.

*Proof.* Similarly to the proof of Theorem 4.4, $\mathcal{J}_f(D)$ rewrites as

$$\mathcal{J}_f(D) = \mathbb{E}_{XY}\Big[D(\mathbf{x},y_{\mathbf{x}})\Big] - \mathbb{E}_X\Big[\sum_{j=1}^{K} f^*(D(\mathbf{x},j))\Big] \tag{74}$$

$$= \sum_{j=1}^{K}\Big(p_Y(j)\mathbb{E}_{X|Y}\Big[D(\mathbf{x},j)\Big] - \mathbb{E}_X\Big[f^*(D(\mathbf{x},j))\Big]\Big) \tag{75}$$

$$= \sum_{j=1}^{K}\mathcal{J}_{f,j}(D) \tag{76}$$

Analogously, for the bias we obtain

$$B_f(D) = \sum_{j=1}^{K}(e_j\mathbb{E}_X\left[D(\mathbf{x},j)\right]) - \left(\sum_{i=1}^{K} e_i\right)\mathbb{E}_X\Big[\sum_{j=1}^{K} f^*(D(\mathbf{x},j))\Big] \tag{77}$$

$$= \sum_{j=1}^{K}\Big(\mathbb{E}_X\Big[e_jD(\mathbf{x},j) - \left(\sum_{i=1}^{K} e_i\right)f^*(D(\mathbf{x},j))\Big]\Big) \tag{78}$$

$$= \sum_{j=1}^{K}B_{f,j}(D). \tag{79}$$

Putting everything together, we obtain

$$\mathcal{J}_f^{\eta}(D) = \left(1 - \sum_{i=1}^{K} e_i\right)\mathcal{J}_f(D) + B_f(D) \tag{80}$$

$$= \left(1 - \sum_{i=1}^{K} e_i\right)\sum_{j=1}^{K}\mathcal{J}_{f,j}(D) + \sum_{j=1}^{K}B_{f,j}(D) \tag{81}$$

$$= \sum_{j=1}^{K}\underbrace{\left(\left(1 - \sum_{i=1}^{K} e_i\right)\mathcal{J}_{f,j}(D) + B_{f,j}(D)\right)}_{\triangleq\mathcal{J}_{f,j}^{\eta}(D)} \tag{82}$$

$$= \sum_{j=1}^{K}\mathcal{J}_{f,j}^{\eta}(D) \tag{83}$$

For the same motivation explained for the binary case, $\mathcal{J}_{f,j}^\eta(D)$ is a concave function of $D$. Therefore, the optimal convergence of $D$ is achieved imposing the first derivative of $\mathcal{J}_{f,j}^\eta(D)$ equal to zero

$$\frac{\partial}{\partial D}\mathcal{J}_{f,j}^\eta(D) = 0 \Rightarrow \tag{84}$$

$$\frac{\partial}{\partial D}\left(\int_{\mathcal{T}_x}\left(1 - \sum_{i=1}^K e_i\right)\left(p_Y(j)p_{X|Y}(\mathbf{x}|j)D(\mathbf{x},j) - p_X(\mathbf{x})f^*(D(\mathbf{x},j))\right) + \tag{85}$$

$$+ p_X(\mathbf{x})e_j D(\mathbf{x},j) - p_X(\mathbf{x})\left(\sum_{i=1}^K e_i\right)f^*(D(\mathbf{x},j))d\mathbf{x}\right) = 0 \tag{86}$$

which implies

$$\left(1 - \sum_{i=1}^K e_i\right)\left(p_Y(j)p_{X|Y}(\mathbf{x}|j) - p_X(\mathbf{x})(f^*)'(D(\mathbf{x},j))\right) + p_X(\mathbf{x})e_j$$

$$- p_X(\mathbf{x})\left(\sum_{i=1}^K e_i\right)(f^*)'(D(\mathbf{x},j)) = 0 \tag{87}$$

$$\Rightarrow \left(1 - \sum_{i=1}^K e_i\right)p_{XY}(\mathbf{x},j) + p_X(\mathbf{x})e_j = p_X(\mathbf{x})(f^*)'(D(\mathbf{x},j)) \tag{88}$$

$$\Rightarrow \left(1 - \sum_{i=1}^K e_i\right)p_{Y|X}(j|\mathbf{x}) + e_j = (f^*)'(D(\mathbf{x},j)). \tag{89}$$

Since $(f^*)'(t) = (f')^{-1}(t)$,

$$D_\eta^\diamond(\mathbf{x},j) = f'\left(\left(1 - \sum_{i=1}^K e_i\right)p_{Y|X}(j|\mathbf{x}) + e_j\right), \tag{90}$$

where $D_\eta^\diamond(\mathbf{x},j)$ is the optimal neural network learned at convergence. Therefore, the posterior estimator obtained in the presence of label noise reads as

$$\hat{p}_{Y|X}^\eta(j|\mathbf{x}) = (f^*)'(D_\eta^\diamond(\mathbf{x},j)) = \left(1 - \sum_{i=1}^K e_i\right)p_{Y|X}(j|\mathbf{x}) + e_j. \tag{91}$$

$\square$

## B.7 SPARSE NOISE

The sparse label noise model characterizes the possibility of having pairwise label flips. This model is extremely useful when two classes in the dataset can be easily confused. For instance, for CIFAR-10, the "cat" and "dog" classes can be more easily confused than "cat" and "truck".

Let $K$ be an even number, there are $K/2$ disjoint pairs of classes $(i_c, j_c)$ where $c \in \left[\frac{K}{2}\right]$ and $i_c < j_c$. Let $\eta_{j_c i_c} = e_0$ and $\eta_{i_c j_c} = e_1$. Let $\eta_{j_c j_c} = 1 - \eta_{j_c i_c} = 1 - e_0$, while $\eta_{i_c i_c} = 1 - \eta_{i_c j_c} = 1 - e_1$.

Assume that $e_0 + e_1 < 1$. Similarly to the proof of Theorem 4.1, the first term can be rewritten as

$$E_{XY_\eta}[D(\mathbf{x}, \tilde{y}_\mathbf{x})] = \mathbb{E}_Y \mathbb{E}_{X|Y} \mathbb{E}_{Y_\eta|Y}[D(\mathbf{x}, \tilde{y}_\mathbf{x})] \tag{92}$$

$$= \sum_{i=1}^K p_Y(i) \mathbb{E}_{X|Y=i} \left[ \sum_{j=1}^K \eta_{ij} D(\mathbf{x}, j) \right] \tag{93}$$

$$= \sum_{i_c} p_{i_c} \mathbb{E}_{X|Y=i_c} [(1-e_1)D(\mathbf{x}, i_c) + e_1 D(\mathbf{x}, j_c)]$$

$$+ \sum_{j_c} p_{j_c} \mathbb{E}_{X|Y=j_c} [(1-e_0)D(\mathbf{x}, j_c) + e_0 D(\mathbf{x}, i_c)] \tag{94}$$

$$= \sum_{i_c} p_{i_c} \mathbb{E}_{X|Y=i_c} [(1-e_0-e_1)D(\mathbf{x}, i_c) + e_1 D(\mathbf{x}, j_c) + e_0 D(\mathbf{x}, i_c)]$$

$$+ \sum_{j_c} p_{j_c} \mathbb{E}_{X|Y=j_c} [(1-e_0-e_1)D(\mathbf{x}, j_c) + e_1 D(\mathbf{x}, j_c) + e_0 D(\mathbf{x}, i_c)] \tag{95}$$

$$= (1-e_0-e_1)\mathbb{E}_{XY}[D(\mathbf{x}, y_\mathbf{x})] + \sum_{(i_c, j_c)} \mathbb{E}_X [e_1 D(\mathbf{x}, j_c) + e_0 D(\mathbf{x}, i_c)] \tag{96}$$

The second term of the objective function is not affected by label noise. Therefore, similarly to the binary case, we can merge the two terms and obtain

$$\mathcal{J}_f^\eta(D) = (1-e_0-e_1)\mathcal{J}_f(D) + B_f(D), \tag{97}$$

where

$$B_f(D) = \left( \sum_{(i_c, j_c)} \mathbb{E}_X [e_0 D(\mathbf{x}, i_c) + e_1 D(\mathbf{x}, j_c)] \right) - (e_0 + e_1)\mathbb{E}_X \left[ \sum_{k=1}^K f^*(D(\mathbf{x}, k)) \right]. \tag{98}$$

Comparing these expressions with Theorem 4.1, it is possible to notice that the sparse case can be treated as $K/2$ disjoint classification problems.

The same consideration can be done for the posterior correction approach, where the prediction obtained by a neural network learned in the presence of label noise can be corrected by considering the binary scenario of the predicted label and the unique one belonging to its pair. For instance, suppose that, for a given sample $\mathbf{x}_i$, the neural network trained in the presence of label noise predicts the class $l$ ($l \in [K]$). Given the already estimated transition probabilities, assume without loss of generality that $l$ flips into $m$ ($m \in [K], m \neq l$) with a certain probability $e_0$ and vice versa $m$ flips into $l$ (with a certain probability $e_1$), then it is possible to correct the predicted class by using Theorem 4.4.

### B.8 PROOF OF THEOREM B.1

**Theorem B.1.** *Let $D_\eta^{(i)}$ be the neural network at the $i$-th step of training maximizing $\mathcal{J}_f^\eta(D)$. Assume $T_\eta^{(i)}$ belongs to the neighborhood of $D_\eta^\diamond$. The bias during training is bounded as*

$$|p_\eta^\diamond - p_\eta^{(i)}| \leq ||(D_\eta^\diamond - D_\eta^{(i)})||_2 ||(f^*)''(D_\eta^{(i)})||_2. \tag{99}$$

*Proof.* The difference between $p_\eta^\diamond$ and $p_\eta^{(i)}$ can be written as

$$p_\eta^\diamond - p_\eta^{(i)} = (f^*)'(D_\eta^\diamond) - (f^*)'(D_\eta^{(i)}) \tag{100}$$

$$\simeq \delta^{(i)}(f^*)''(D_\eta^{(i)}) \tag{101}$$

$$= (D_\eta^\diamond - D_\eta^{(i)})(f^*)''(D_\eta^{(i)}) \tag{102}$$

Thus,

$$|p_\eta^\diamond - p_\eta^{(i)}| = |(D_\eta^\diamond - D_\eta^{(i)})(f^*)''(D_\eta^{(i)})| \leq ||(D_\eta^\diamond - D_\eta^{(i)})||_2 ||(f^*)''(D_\eta^{(i)})||_2 \tag{103}$$

for the Cauchy-Schwarz inequality. $\square$

## B.9 Proof of Theorem B.2

**Theorem B.2.** *Let $D_{\eta j}^{\diamond}$ and $D_{\eta j}^{(i)}$ the $j$-th output of the posterior estimator at convergence and at the $i$-th iteration of training, respectively. The difference between the optimal posterior estimate without label noise and the estimate at $i$-th iteration in the presence of label noise reads as*

$$p_j^{\diamond} - p_{\eta j}^{(i)} \simeq \left(\sum_{n=1}^{K} e_n\right) p_j^{\diamond} - e_j + \delta_j^{(i)}(f^*)''(D_{\eta j}^{\diamond} - \delta_j^{(i)}), \tag{104}$$

*where $\delta_j^{(i)} = D_{\eta j}^{\diamond} - D_{\eta j}^{(i)}$.*

*Proof.* We can study the bias of the estimator during training as

$$p^{\diamond} - p_{\eta}^{(i)} = (f^*)'(D^{\diamond}) - (f^*)'(D_{\eta}^{(i)}) \tag{105}$$

$$= (f^*)'(D^{\diamond}) - (f^*)'(D_{\eta}^{\diamond} - \delta^{(i)}) \tag{106}$$

$$\simeq (f^*)'(D^{\diamond}) - (f^*)'(D_{\eta}^{\diamond}) + \delta^{(i)}(f^*)''(D_{\eta}^{\diamond} - \delta^{(i)}) \tag{107}$$

where the last step is obtained using the Taylor expansion. In the binary case, for the $j$-th class, we get

$$p_j^{\diamond} - p_{\eta j}^{(i)} \simeq (f^*)'(D_j^{\diamond}) - [(1 - e_0 - e_1)(f^*)'(D_j^{\diamond}) + e_j] + \delta_j^{(i)}(f^*)''(D_{\eta j}^{\diamond} - \delta_j^{(i)}) \tag{108}$$

$$= (f^*)'(D_j^{\diamond})[1 - (1 - e_0 - e_1)] - e_j + \delta_j^{(i)}(f^*)''(D_{\eta j}^{\diamond} - \delta_j^{(i)}) \tag{109}$$

$$= [e_0 + e_1](f^*)'(D_j^{\diamond}) - e_j + \delta_j^{(i)}(f^*)''(D_{\eta j}^{\diamond} - \delta_j^{(i)}) \tag{110}$$

$$= [e_0 + e_1]p_j^{\diamond} - e_j + \delta_j^{(i)}(f^*)''(D_{\eta j}^{\diamond} - \delta_j^{(i)}). \tag{111}$$

In the multi-class case, for the $j$-th output of the discriminator, we get

$$p_j^{\diamond} - p_{\eta j}^{(i)} \simeq (f^*)'(D_j^{\diamond}) - [(1 - \sum_{i=1}^{K} e_i)(f^*)'(D_j^{\diamond}) + e_j] + \delta_j^{(i)}(f^*)''(D_{\eta j}^{\diamond} - \delta_j^{(i)}) \tag{112}$$

$$= \left(\sum_{i=1}^{K} e_i\right)(f^*)'(D_j^{\diamond}) - e_j + \delta_j^{(i)}(f^*)''(D_{\eta j}^{\diamond} - \delta_j^{(i)}) \tag{113}$$

$$= \left(\sum_{i=1}^{K} e_i\right)p_j^{\diamond} - e_j + \delta_j^{(i)}(f^*)''(D_{\eta j}^{\diamond} - \delta_j^{(i)}). \tag{114}$$

$\square$

## C  Comparison with Related Work

**Robust objective functions** These algorithms utilize objective functions inherently robust to label noise. In Menon et al. (2015), the authors demonstrate robustness of deep learning algorithms for binary classification under symmetric label noise. In Ghosh et al. (2017), the authors prove the robustness of symmetric objective functions. In particular, they show that the CE is not symmetric, while proving that the mean absolute error (MAE) is a robust loss. In Zhang & Sabuncu (2018), the authors show that MAE performs poorly for challenging datasets and propose the generalized cross-entropy (GCE), which is a trade-off between MAE and categorical CE, leveraging the negative Box-Cox transformation. Symmetric Cross Entropy (SCE) Wang et al. (2019) combines the CE loss with a Reverse Cross Entropy (RCE) loss robust to label noise, to avoid overfitting to noisy labels. In Xu et al. (2019), the authors propose a robust loss function based on the determinant-based mutual information. In Ma et al. (2020), the authors prove that all the objective functions can be made robust to label noise with a normalization. However, they show that robust losses can suffer from an underfitting issue. Therefore, they propose a class of objective functions, referred to as active passive losses (APLs), that mitigate the underfitting problem. Peer Loss functions Liu & Guo (2020) are a class of robust loss functions inspired by correlated agreement. In Wei & Liu (2021), the

authors propose a class of objective functions based on the maximization of the $f$-divergence-based generalization of mutual information. In Ye et al. (2023), the authors propose a specific class of APLs, referred to as active negative loss functions (ANLs), that, instead of obtaining the passive losses based on MAE as in Ma et al. (2020), use negative loss functions based on complementary label learning Ishida et al. (2017). In Zhou et al. (2023), the authors propose a class of loss functions robust to label noise that extend symmetric losses, while stressing the urgency of designing non-symmetric objective functions robust to label noise. In Zhu et al. (2024), the authors prove the robustness of deep learning algorithms to instance-dependent noise when the transition matrix is strictly diagonally dominant.

## C.1 ACTIVE PASSIVE LOSSES

In this section, we first recall the definitions of active and passive losses from Ma et al. (2020). Then, we show that the class of objective functions in equation 6 is composed by the sum of an active and a passive objective functions.

**Definition C.1** (Active loss function (see Ma et al. (2020))). $\mathcal{J}_{Active}$ is an active loss function if $\forall (\mathbf{x}, \mathbf{y_x}) \in \mathcal{D}, \forall k \neq \mathbf{y_x} \, l(f(\mathbf{x}), k) = 0$.

**Definition C.2** (Passive loss function (see Ma et al. (2020))). $\mathcal{J}_{Passive}$ is a passive loss function if $\forall (\mathbf{x}, \mathbf{y_x}) \in \mathcal{D}, \exists k \neq \mathbf{y_x}$ such that $l(f(\mathbf{x}), k) \neq 0$.

Definition C.1 describes objective functions that are only affected by the prediction corresponding to the label. All the predictions corresponding to a class different from the label of the sample $\mathbf{x}$ are irrelevant. Definition C.2 describes objective functions for which at least one of the neural network's predictions corresponding to a class different from the label contributes to the objective function value.

Following definitions C.1 and C.2, the class of objective functions in equation 6 can be rewritten as $\mathcal{J}_f = \mathcal{J}_{Active} + \mathcal{J}_{Passive}$, where $\mathbb{E}_{XY}[\cdot]$ is the active term, and $\mathbb{E}_X[\cdot]$ is the passive term.

In Ye et al. (2023), the authors study the APLs proposed in Ma et al. (2020) and notice that the passive losses proposed in Ma et al. (2020) are all scaled versions of MAE. Therefore, they propose a new class of passive loss functions based on complementary label learning and vertical flipping. They show that this new class of passive losses performs better than the one used in Ma et al. (2020).

Differently from Ye et al. (2023), in this paper the active and passive objective functions are jointly related to the $f$-divergence used. In other words, the passive term depends on the active and vice versa.

## C.2 $f$-DIVERGENCE FOR NOISY LABELS

The $f$-divergence has been used in learning with noisy labels in Wei & Liu (2021), where the authors maximize the $f$-MI (which is a generalization of the mutual information (MI)) between the label distribution and the classifier's output distribution. Let $X \in \mathcal{X}$ and $Y \in \mathcal{Y}$ be two random vectors having probability density functions $p_X(\mathbf{x})$ and $p_Y(\mathbf{y})$, respectively. Let $y_\mathbf{x}$ be the label corresponding to an object $\mathbf{x}$ (e.g., an image), the MI between $X$ and $Y$ is defined as

$$I(X;Y) = \mathbb{E}_{XY}\left[\underbrace{\log\left(\frac{p_{XY}(\mathbf{x}, y_\mathbf{x})}{p_X(\mathbf{x})p_Y(y_\mathbf{x})}\right)}_{\triangleq \iota(\mathbf{x}; y_\mathbf{x})}\right], \tag{115}$$

where $\iota(\mathbf{x}; y_\mathbf{x})$ is the pointwise mutual information (PMI).

Several machine learning approaches rely on the maximization of MI, for instance for representation learning Hjelm et al. (2019) and communication engineering Letizia et al. (2023) applications. However, the maximization of MI does not always lead to learning the best models, as showed in Tschannen et al. (2020) for the representation learning domain. In this specific scenario, there is no guarantee that the maximization of the $f$-MI is a classification objective which leads to a Bayes classifier

$$C^B(\mathbf{x}) = \underset{i \in \{1, \ldots, K\}}{\arg\max} \, P(Y = i | X = \mathbf{x}). \tag{116}$$

The authors of Wei & Liu (2021), in fact, proved that in the binary classification scenario maximizing the $f$-MI leads to the Bayes optimal classifier only when the classes in the dataset have equal prior probability (i.e., it is a balanced dataset) and when using a restricted set of $f$-divergences (e.g., the total variation). They extend their findings for the multi-class scenario only for confident classifiers.

Differently, let $\mathcal{A}_y$ be a set of $K$ classes, maximizing the PMI corresponds to finding the optimal Bayes classifier, i.e.,

$$\hat{y}_\mathbf{x} = \arg\max_{y_\mathbf{x} \in \mathcal{A}_y} \iota(\mathbf{x}; y_\mathbf{x}) = \arg\max_{y_\mathbf{x} \in \mathcal{A}_y} p_{Y|X}(y_\mathbf{x}|\mathbf{x}), \tag{117}$$

since $p_Y$ is fixed given a dataset. To compare $f$-NPL and the approach in Wei & Liu (2021), we notice that $f$-NPL relies on the maximization of **PMI** between objects (e.g., images), thus returning the Bayes optimal classifier (equation 116) for *any* $f$-divergence, by definition.

In addition, variational MI estimators are upper bounded McAllester & Stratos (2020). The main reason is that they need to draw samples from $p_X(\mathbf{x})p_Y(y)$. However, practically it is difficult to ensure that, given a batch of samples drawn from $p_{XY}(\mathbf{x}, y_\mathbf{x})$, a random shuffle/derangement of the batch of $y$ returns a batch of samples from $p_X(\mathbf{x})p_Y(y)$. This is still an open problem McAllester & Stratos (2020); Letizia et al. (2024) which bounds MI estimates of discriminative estimators. Differently, $f$-NPL does not need to break the relationship between the realizations of $X$ and $Y$ through a shuffling mechanism to draw the samples from $p_X(\mathbf{x})p_Y(y)$, because it only needs samples from $p_{XY}(\mathbf{x}, y_\mathbf{x})$.

Finally, the objective function in Wei & Liu (2021) is robust to symmetric and asymmetric off-diagonal label noise (class conditional label noise) for a *restricted* class of $f$-divergences, while $f$-NPL is intrinsically robust to instance-dependent label noise for *any* $f$-divergence (under some assumptions on the transition matrix).

## D ADDITIONAL EXPERIMENTAL RESULTS

### D.1 IMPLEMENTATION DETAILS

**Datasets description** For the binary classification scenario, we use the breast cancer dataset Wolberg et al. (1993) available on Scikit-learn Pedregosa et al. (2011). It contains 569 samples and 30 features. For the multiclass classification task, we use datasets with synthetic label noise generated from CIFAR-10 and CIFAR-100 Krizhevsky et al. (2009). These consist of $60k$ $32 \times 32$ images split in $50k$ for training and $10k$ for test. CIFAR-10 contains 10 classes, with 6000 images per class. CIFAR-100 contains 100 classes, with 600 images per class. Following previous work, the synthetic symmetric label noise is generated by randomly flipping the label of a given percentage of samples into a fake label with a uniform probability, while the asymmetric label noise is generated by flipping labels for specific classes. For datasets with realistic label noise, we use CIFAR-10N and CIFAR-100N Wei et al. (2021). CIFAR-10N contains human annotations from three independent workers (Random 1, Random 2, and Random 3) which are combined by majority voting to get an aggregated label (Aggregate) and to get wrong labels (Worst). CIFAR-100N contains human annotations submitted for the fine classes. For mini WebVision Li et al. (2017) and Imagenet ILSVRC12 Krizhevsky et al. (2012) we use the datasets with original labels. Indeed, they are large-scale image datasets that already contain label noise. For instance, WebVision contains approximately $20\%$ of samples with wrong labels Li et al. (2017).

**Hyperparameters and network architecture** The correction techniques were evaluated by running from scratch the experiments for all the methods (i.e., Forward loss, RENT, and $f$-NPL). For CIFAR-10, we use a ResNet34, while for CIFAR-100 we use a ResNet50. For the remaining experiments in this paper, we report the results of the various techniques compared with $f$-NPL as reported in the corresponding papers. For the comparisons with APL-like objective functions on CIFAR-10 and CIFAR-10N, we use the same 8-layer CNN used in Ma et al. (2020); Ye et al. (2023). Meanwhile, for the comparison with APL-like objective functions on CIFAR-100 and CIFAR-100N, we use a ResNet34 He et al. (2016). For the experiments on large real-world datasets, we train a ResNet50 on mini WebVision. For $f$-NPL$_{\text{Pro}}$ (i.e., $f$-NPL employing the ProMix Wang et al. (2022) architecture),

we use the ProMix architecture of the original paper, consisting of 2 ResNet18. For the ProMix training strategy, we use the same hyperparameters reported in Wang et al. (2022)[1].

Optimization is executed using SGD with a momentum of 0.9. The learning rate is initially set to 0.02 and a cosine annealing scheduler Loshchilov & Hutter (2017) decays it during training. We decided to keep the learning rate 0.02 for $f$-NPL and the other methods, while maintaining 0.01 and 0.1 for the APL-like losses so that each algorithm is trained with the optimal initial learning rate for that specific algorithm. Forcing different methods to use the same, potentially suboptimal, learning rate, would unjustly penalize one of them. For the experiments about the correction techniques, we train the algorithms for 200 epochs with a batch size of 128. For the experiments on the binary dataset, we trained the models for 100 epochs, with a batch size of 32. For the comparison with APL-like losses on the CIFAR-10 and CIFAR-10N datasets, we trained the neural networks for 120 epochs, with a batch size of 128. For the comparison with APL-like losses on the CIFAR-100 and CIFAR-100N datasets, we trained the neural networks for 200 epochs, with a batch size of 128. For mini WebVision, the training lasts for 100 epochs and we use a batch size of 64. For $f$-NPL$_{Pro}$, the training lasts for 600 epochs, with a batch size of 256. All the tables report the mean over 5 independent runs of the code with different random seeds. Some also report the standard deviation. The experiments are run on a server with CPU "AMD Ryzen Threadripper 3960X 24-Core Processor" and GPU "MSI GeForce RTX 3090 Gaming X Trio 24G, 24GB GDDR6X".

**Baselines** All the baselines are reported in the following: standard cross-entropy minimization approach (CE), Forward loss (FL) Patrini et al. (2017), GCE Zhang & Sabuncu (2018), Co-teaching Han et al. (2018), Co-teaching+ Yu et al. (2019), SCE Wang et al. (2019), NLNL Kim et al. (2019), JoCoR Wei et al. (2020), ELR Liu et al. (2020), Peer Loss Liu & Guo (2020), NCE+RCE/NCE+MAE/NFL+RCE/NFL+MAE Ma et al. (2020), NCE+AEL/NCE+AGCE/NCE+AUL Zhou et al. (2021), F-Div Wei & Liu (2021), Divide-Mix Li et al. (2020), Negative-LS Wei et al. (2022), CORES[2] Cheng et al. (2021), SOP Liu et al. (2022), ProMix Wang et al. (2022), ANL-CE/ANL-FL Ye et al. (2023), RDA Lienen & Hüllermeier (2024), SGN Englesson & Azizpour (2024), RENT Bae et al. (2024). For the feature extraction methods, we use DINOv2 Oquab et al. (2023) and different ResNet He et al. (2016) models available on PyTorch Paszke et al. (2019), pre-trained using self-supervised learning on ImageNet Krizhevsky et al. (2012).

### D.2 ADDITIONAL RESULTS

#### D.2.1 OBJECTIVE FUNCTION AND POSTERIOR CORRECTION

**Sparse label noise** Additionally to the symmetric and asymmetric label noise types studied in Tab. 1, we evaluate the correction approaches for the case of sparse label noise in Tab. 4. As demonstrated by Tab. 1, $f$-NPL performs better than the other correction approaches for different label noise scenarios and different methods to estimate the transition matrix.

Table 4: Test accuracy comparison for sparse label noise on CIFAR-10 and CIFAR-100.

| T Est. | Method | CIFAR-10 | | | CIFAR-100 | | |
|---|---|---|---|---|---|---|---|
| | | SpN 10% | SpN 20% | SpN 30% | SpN 10% | SpN 20% | SpN 30% |
| Forward | w/FL | $90.39_{\pm0.3}$ | $89.98_{\pm0.1}$ | $88.52_{\pm0.8}$ | $65.75_{\pm0.2}$ | $64.29_{\pm0.3}$ | $59.16_{\pm0.6}$ |
| | w/RENT | $88.51_{\pm0.7}$ | $88.06_{\pm0.9}$ | $86.73_{\pm1.2}$ | $62.62_{\pm0.9}$ | $61.08_{\pm0.8}$ | $57.19_{\pm1.1}$ |
| | w/$f$-NPL$_o$ | $93.68_{\pm0.2}$ | $\mathbf{92.43_{\pm0.4}}$ | $\mathbf{89.50_{\pm0.5}}$ | $76.06_{\pm0.3}$ | $\mathbf{72.80_{\pm0.6}}$ | $\mathbf{68.34_{\pm0.8}}$ |
| | w/$f$-NPL$_p$ | $\mathbf{93.68_{\pm0.1}}$ | $91.70_{\pm0.3}$ | $88.00_{\pm0.4}$ | $\mathbf{76.30_{\pm0.7}}$ | $72.34_{\pm0.6}$ | $64.72_{\pm0.7}$ |
| DualT | w/FL | $90.40_{\pm0.2}$ | $90.01_{\pm0.7}$ | $88.61_{\pm0.4}$ | $67.04_{\pm0.3}$ | $65.64_{\pm0.5}$ | $52.82_{\pm0.6}$ |
| | w/RENT | $87.91_{\pm0.5}$ | $86.87_{\pm0.9}$ | $85.40_{\pm0.8}$ | $63.25_{\pm1.2}$ | $59.27_{\pm1.4}$ | $56.77_{\pm1.1}$ |
| | w/$f$-NPL$_o$ | $93.30_{\pm0.2}$ | $91.15_{\pm0.3}$ | $89.78_{\pm0.6}$ | $76.23_{\pm0.7}$ | $74.21_{\pm0.5}$ | $67.10_{\pm0.8}$ |
| | w/$f$-NPL$_p$ | $\mathbf{93.72_{\pm0.4}}$ | $\mathbf{91.75_{\pm0.4}}$ | $\mathbf{90.06_{\pm0.6}}$ | $\mathbf{77.09_{\pm0.5}}$ | $\mathbf{72.81_{\pm0.8}}$ | $\mathbf{68.95_{\pm0.9}}$ |
| True $T$ | w/FL | $90.54_{\pm0.2}$ | $88.37_{\pm0.3}$ | $87.41_{\pm0.3}$ | $68.26_{\pm0.2}$ | $68.10_{\pm0.9}$ | $\mathbf{67.70_{\pm0.7}}$ |
| | w/RENT | $88.75_{\pm0.6}$ | $87.66_{\pm0.8}$ | $86.80_{\pm0.9}$ | $64.02_{\pm0.8}$ | $62.35_{\pm1.2}$ | $61.35_{\pm1.1}$ |
| | w/$f$-NPL$_o$ | $93.31_{\pm0.3}$ | $\mathbf{91.79_{\pm0.3}}$ | $87.79_{\pm0.4}$ | $\mathbf{75.96_{\pm0.2}}$ | $70.61_{\pm0.5}$ | $64.38_{\pm0.6}$ |
| | w/$f$-NPL$_p$ | $\mathbf{93.89_{\pm0.2}}$ | $91.75_{\pm0.3}$ | $\mathbf{88.31_{\pm0.5}}$ | $75.61_{\pm0.4}$ | $\mathbf{72.64_{\pm0.7}}$ | $66.42_{\pm0.8}$ |
| No Corr. | $f$-**NPL** | $93.51_{\pm0.3}$ | $91.84_{\pm0.4}$ | $87.78_{\pm0.4}$ | $75.63_{\pm0.5}$ | $72.71_{\pm0.7}$ | $66.61_{\pm0.6}$ |

[1]See the GitHub repository of ProMix https://github.com/Justherozen/ProMix

Table 5: Test accuracy comparison on breast cancer test dataset for $[e_0, e_1] = [0.1, 0.3]$.

| Div. | No Cor. | O.F. Cor. | P. Cor. | No Noise |
|------|---------|-----------|---------|----------|
| KL-NPL | 92.10 | 95.60 | 95.60 | 98.20 |
| SL-NPL | 92.10 | 95.60 | 95.60 | 98.20 |
| GAN-NPL | 93.00 | 94.70 | 95.60 | 98.20 |

Table 6: Test accuracy comparison on breast cancer test dataset for $[e_0, e_1] = [0.2, 0.4]$.

| Div. | No Cor. | O.F. Cor. | P. Cor. | No Noise |
|------|---------|-----------|---------|----------|
| KL-NPL | 90.40 | 94.70 | 92.20 | 98.20 |
| SL-NPL | 87.70 | 93.90 | 91.30 | 98.20 |
| JS-NPL | 89.00 | 94.70 | 92.20 | 98.20 |

**Binary classification** We evaluate the binary objective function and posterior correction approaches on the breast cancer classification dataset Wolberg et al. (1993) available on Scikit-learn Pedregosa et al. (2011). We study the performance of the various correction approaches for different values of the noise rates and divergences. For binary datasets, the transition matrix becomes

$$T = \begin{bmatrix} 1 - e_1 & e_1 \\ e_0 & 1 - e_0 \end{bmatrix}. \tag{118}$$

The test accuracy comparison between KL-NPL, SL-NPL, and GAN-NPL using the objective function correction (O.F. Cor.) and posterior correction (P. Cor.) approaches is reported in Tabs. 5 and 6 for $e_0 = 0.1$, $e_1 = 0.3$ and $e_0 = 0.2$, $e_1 = 0.4$, respectively.

**Additional analysis of correction approaches** We plot the test accuracy attained during training for different correction techniques using the true transition matrix, for CIFAR-10 with asymmetric label noise and $\eta = 0.4$ in Fig. 4, showing mean and standard deviation obtained over multiple random seeds. $f$-NPL$_o$ and $f$-NPL$_p$ indicate the usage of $f$-NPL with objective function correction and posterior correction, respectively. Fig. 4 shows the gap between $f$-NPL and the other approaches, both in terms of accuracy and variance. Notably, the convex shape of the accuracy curve in the early stage of training derives from the usage of a pre-training stage. In fact, following the work in Bae et al. (2024), the training is preceded by a pre-training step of a maximum of 20 epochs with CE, in which the transition matrix is estimated. Without pre-training, the curve has a monotonic-like behavior (see Fig. 6). The first column of Fig. 5 shows the test accuracy attained by $f$-NPL (first row), $f$-NPL$_o$ (second row), and $f$-NPL$_p$ (last row) varying the noise rate, for asymmetric label noise on CIFAR-10, using the true transition matrix for corrections. In addition to the test accuracy, we report the accuracy on the training dataset with noisy labels (central column) and the original clean labels (last column). From these plots, it is possible to formulate some observations regarding the memorization phenomenon, which is an important topic in learning with noisy labels Arpit et al. (2017). Fig. 5 shows that, in this specific scenario, with a training that lasts 200 epochs, the test and training accuracy of $f$-NPL$_o$ increase during training, which is a symptom of no memorization. The same holds true for medium and low values of the noise rate for $f$-NPL and $f$-NPL$_p$. However, for high noise rates, it seems that in the final stage of training the test accuracy decreases while the training accuracy increases, indicating that they could be more subject to memorization than $f$-NPL$_o$.

To demonstrate that the peculiar early-training convexity of Fig. 5 is due to the pre-training performed with the CE, we report in Fig. 6 the test accuracy for different noise rates on CIFAR-10 with asymmetric label noise, without pre-training. Also in Fig. 6, as in Fig. 5, we use the true transition matrix for performing correction. In Fig. 6, the convex-like shape of the test accuracy in the early phase of training disappears, showing a behavior that is close to a monotonically increasing function. However, $f$-NPL$_o$ performs significantly worse for few specific high noise rates, indicating the necessity of performing pre-training when using the objective function correction. By comparing Fig. 6 with Fig. 5 and Tab. 1, we can notice that, in this specific setting: 1) FL and RENT achieve a higher accuracy when the pre-training is performed. 2) $f$-NPL$_o$ benefits from the pre-training for all the noise rates, in fact, without pre-training it achieves a lower maximum accuracy also for low noise rates. 3) $f$-NPL is not strongly affected by the pre-training. 4) $f$-NPL$_p$ performs better without pre-training. In addition, it appears that the test accuracy difference between the early stage and the

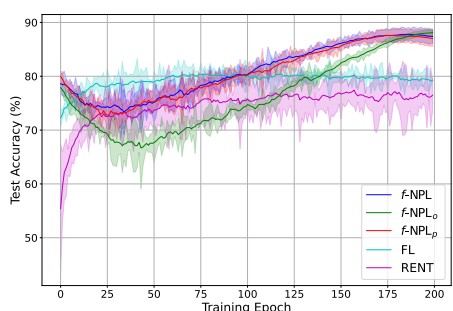

Figure 4: Mean (saturated) $\pm$ standard deviation (transparent) test accuracy over training epochs.

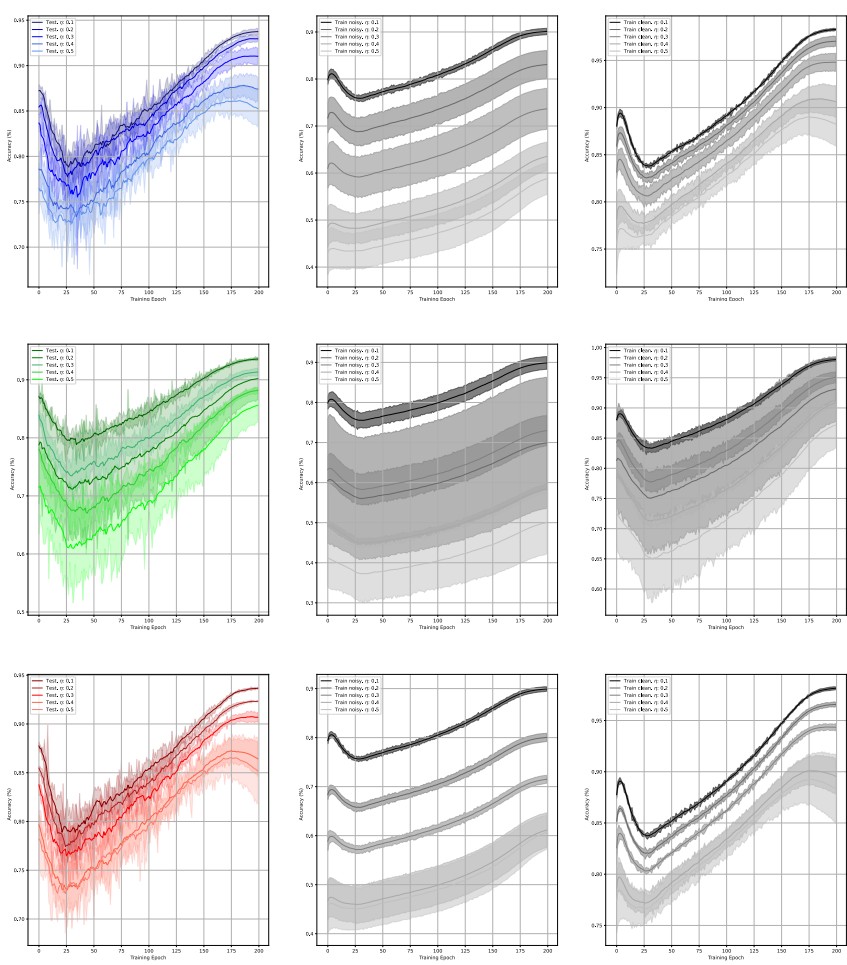

Figure 5: Test accuracy (left column), training accuracy on noisy labels (central column), and training accuracy on original clean labels (right column) for $f$-NPL (top row), $f$-NPL$_o$ (central row), and $f$-NPL$_p$ (bottom row) on CIFAR-10 with asymmetric label noise. Different colors correspond to different noise rates.

end of training is relatively small for FL and RENT, while it is larger for $f$-NPL (with and without correction).

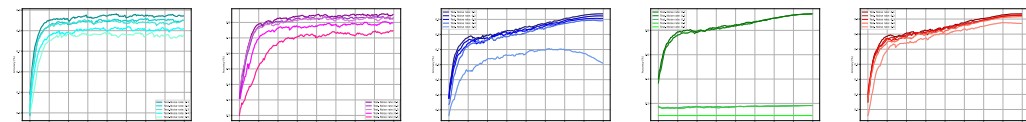

Figure 6: Test accuracy on CIFAR-10 with asymmetric label noise without pre-training with CE. From left to right: FL, RENT, $f$-NPL, $f$-NPL$_o$, and $f$-NPL$_p$.

**Effect of the inaccurate transition matrix estimation**  When the transition matrix is slightly inaccurately estimated (for instance when it is estimated using Forward correction and DualT), the test accuracy of $f$-NPL remains higher than the accuracy obtained from the other correction methods, as already demonstrated in Tabs. 1 and 4. To investigate the performance of the algorithms in a systematic way, we study a scenario where we gradually increase the inaccuracy of the transition matrix estimation. We consider the case of asymmetric label noise and we set the true noise rate to 20%, since in real-world scenarios the noise rate is usually between 8% and 38.5% Song et al. (2022); Xiao et al. (2015); Li et al. (2017); Lee et al. (2018); Song et al. (2019). Then, we study the case in which the noise rate of the estimated transition matrix $\hat{T}$ (referred to as $\hat{\eta}$) is higher than the noise rate of the true $T$ (referred to as $\eta$) by a parameter $\delta_\eta$ (i.e., $\hat{\eta} = \eta + \delta_\eta$). We gradually vary $\delta_\eta$ from 0.2 to 0.4, studying its effect on the test accuracy, and we report the results in Tab. 7.

Table 7: Test accuracy on CIFAR-10 with asymmetric noise when the transition matrix is inaccurately estimated. All methods use ResNet34. Each value is the mean obtained over three random seeds.

| Method | $\delta_\eta = 0.2$ | $\delta_\eta = 0.3$ | $\delta_\eta = 0.4$ |
|---|---|---|---|
| FL | 84.81 | 84.25 | 82.72 |
| RENT | 85.52 | 82.53 | 81.58 |
| $f$-NPL$_o$ | 91.16 | 90.53 | 87.88 |
| $f$-NPL$_p$ | **92.61** | **92.53** | **92.35** |

### D.2.2  ADDITIONAL COMPARISON WITH ROBUST METHODS

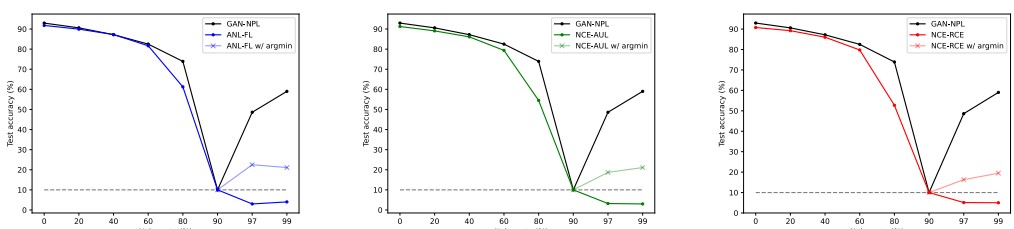

Figure 7: Comparison of GAN-NPL (in black) with three different APL-like losses (with opaque colors). For each APL-like loss, for noise rates that violate the robustness condition, we also plot the test accuracy obtained using the argmin instead of the standard argmax (with blended colors) to estimate the samples' class, even if there is no theoretical guarantee about their robustness in such situations, differently from $f$-NPL.

We report a comparison of the test accuracy achieved from GAN-NPL and three APL-like losses (ANL-FL Ye et al. (2023), NCE-AUL Zhou et al. (2023), NCE-RCE Ma et al. (2020)) varying the noise rate in Fig. 7, for CIFAR-10 with symmetric label noise. This comparison further validates the theoretical results presented in Sec. 3. For all the algorithms, when the noise rate increases from 0% to 90%, the test accuracy decreases, which is a standard behavior even if they are all robust to symmetric label noise, as the estimated posterior gradually shifts towards a uniform probability, thus having a smaller gap between the probability of the true label and the probability of all the other labels (see an example in Fig. 8). When the noise rate is exactly 90%, the noisy posterior corresponds to a vector with all elements equal to $1/K$, and therefore the test accuracy coincides with 10%, which

$$p_{Y|X} = \begin{bmatrix} 1 \\ 0 \\ \vdots \\ 0 \end{bmatrix} \qquad \eta = 0.2 \atop p_{Y_\eta|X} = \begin{bmatrix} 0.800 \\ 0.022 \\ \vdots \\ 0.022 \end{bmatrix} \qquad \eta = 0.9 \atop p_{Y_\eta|X} = \begin{bmatrix} 0.1 \\ 0.1 \\ \vdots \\ 0.1 \end{bmatrix} \qquad \eta = 0.97 \atop p_{Y_\eta|X} = \begin{bmatrix} 0.030 \\ 0.108 \\ \vdots \\ 0.108 \end{bmatrix} \qquad \eta = 0.999 \atop p_{Y_\eta|X} = \begin{bmatrix} 0.001 \\ 0.111 \\ \vdots \\ 0.111 \end{bmatrix}$$

Increasing $\eta$ →

Figure 8: Example of symmetric label noise effect on the true posterior, for CIFAR-10. When the noise rate moves away from the threshold $\frac{K-1}{K} = 0.9$, the gap between the value of the noisy posterior corresponding to the true label and the other labels increases, thus making it easier for $f$-NPL to correctly classify a sample.

is a theoretical limitation that cannot be improved and which coincides with random guessing. The interesting part of Fig. 7 is the one related to noise rates higher than 90%. When the noise rate exceeds 90%, GAN-NPL predicts the class using the argmin instead of the argmax, while all the other APL-like losses continue using the argmax (whose prediction is highlighted by the opaque plots in Fig. 7). GAN-NPL significantly outperforms the other APL-like losses. For curiosity, we also tried to use the argmin operation (reported in Fig. 7 with blended colors) for the other APL-like losses for extremely high noise rates, even if *this does not have any theoretical guarantee of correctness*: GAN-NPL still outperforms the other APL-like losses. Fig. 2 condenses the information of the three plots of Fig. 7: the orange area is limited by the minimum and maximum test accuracies achieved by the other APL-like methods, for each noise rate.

Tab. 8 shows that $f$-NPL outperforms other APL-like losses for symmetric label noise on CIFAR-10 and CIFAR-100.

Table 8: Test accuracy of methods with an APL-like objective function in the presence symmetric label noise, using an 8-layer CNN for CIFAR-10, and a ResNet34 for CIFAR-100.

| Method | CIFAR-10 | | | | | CIFAR-100 | |
|---|---|---|---|---|---|---|---|
| | Clean | 20% | 40% | 60% | 80% | Clean | 20% |
| NFL+MAE | $89.25_{\pm0.19}$ | $87.33_{\pm0.14}$ | $83.81_{\pm0.06}$ | $76.36_{\pm0.31}$ | $45.23_{\pm0.52}$ | $67.98_{\pm0.52}$ | $63.58_{\pm0.09}$ |
| NFL+RCE | $90.91_{\pm0.02}$ | $89.14_{\pm0.13}$ | $86.05_{\pm0.12}$ | $79.78_{\pm0.13}$ | $55.06_{\pm1.08}$ | $68.23_{\pm0.62}$ | $64.52_{\pm0.35}$ |
| NCE+MAE | $88.83_{\pm0.34}$ | $87.12_{\pm0.21}$ | $84.19_{\pm0.43}$ | $77.61_{\pm0.05}$ | $49.62_{\pm0.72}$ | $68.75_{\pm0.54}$ | $65.25_{\pm0.62}$ |
| NCE+RCE | $90.76_{\pm0.22}$ | $89.22_{\pm0.27}$ | $86.02_{\pm0.09}$ | $79.78_{\pm0.50}$ | $52.71_{\pm1.90}$ | $69.02_{\pm0.11}$ | $65.31_{\pm0.07}$ |
| NCE+AEL | $88.51_{\pm0.26}$ | $86.59_{\pm0.24}$ | $83.07_{\pm0.46}$ | $75.06_{\pm0.26}$ | $41.79_{\pm1.40}$ | $64.98_{\pm0.42}$ | $48.13_{\pm0.32}$ |
| NCE+AGCE | $91.08_{\pm0.06}$ | $89.11_{\pm0.07}$ | $86.16_{\pm0.10}$ | $80.14_{\pm0.27}$ | $55.62_{\pm4.78}$ | $68.61_{\pm0.12}$ | $65.30_{\pm0.21}$ |
| NCE+AUL | $91.26_{\pm0.12}$ | $89.08_{\pm0.14}$ | $86.11_{\pm0.27}$ | $79.39_{\pm0.41}$ | $54.49_{\pm2.77}$ | $69.91_{\pm0.18}$ | $65.26_{\pm0.17}$ |
| ANL-CE | $91.66_{\pm0.04}$ | $90.02_{\pm0.23}$ | $87.28_{\pm0.02}$ | $81.12_{\pm0.30}$ | $61.27_{\pm0.55}$ | $70.68_{\pm0.23}$ | $\mathbf{66.79}_{\pm0.34}$ |
| ANL-FL | $91.79_{\pm0.19}$ | $89.95_{\pm0.20}$ | $87.25_{\pm0.11}$ | $81.67_{\pm0.19}$ | $61.22_{\pm0.85}$ | $70.40_{\pm0.15}$ | $66.54_{\pm0.29}$ |
| **SL-NPL** | $92.75_{\pm0.15}$ | $\mathbf{91.16}_{\pm0.21}$ | $\mathbf{87.44}_{\pm0.19}$ | $81.85_{\pm0.28}$ | $64.27_{\pm0.61}$ | $77.58_{\pm0.23}$ | $66.62_{\pm0.35}$ |
| **GAN-NPL** | $\mathbf{92.96}_{\pm0.09}$ | $90.59_{\pm0.16}$ | $87.20_{\pm0.18}$ | $\mathbf{82.51}_{\pm0.23}$ | $\mathbf{73.91}_{\pm0.56}$ | $\mathbf{78.91}_{\pm0.17}$ | $65.33_{\pm0.42}$ |

We show that $f$-NPL is also competitive with well-known algorithms for classification with label noise that do not use APL-like objective functions in Tab. 9, for symmetric label noise.

Table 9: Test accuracy on CIFAR-10 with symmetric noise. All methods use ResNet34.

| Method | Symmetric | | | |
|---|---|---|---|---|
| | 20% | 40% | 60% | 80% |
| CE | $86.32_{\pm0.18}$ | $82.65_{\pm0.16}$ | $76.15_{\pm0.32}$ | $59.28_{\pm0.97}$ |
| GCE | $89.83_{\pm0.20}$ | $87.13_{\pm0.22}$ | $82.54_{\pm0.23}$ | $64.07_{\pm1.38}$ |
| SCE | $87.86_{\pm0.12}$ | $79.96_{\pm0.25}$ | $62.16_{\pm0.33}$ | $27.98_{\pm0.98}$ |
| ELR | $91.16_{\pm0.08}$ | $89.15_{\pm0.17}$ | $86.12_{\pm0.49}$ | $73.86_{\pm0.61}$ |
| SOP | $93.18_{\pm0.57}$ | $90.09_{\pm0.27}$ | $\mathbf{86.76}_{\pm0.22}$ | $68.32_{\pm0.77}$ |
| SL-NPL | $92.97_{\pm0.37}$ | $\mathbf{90.38}_{\pm0.41}$ | $85.25_{\pm0.44}$ | $65.29_{\pm0.86}$ |
| GAN-NPL | $\mathbf{93.20}_{\pm0.13}$ | $90.05_{\pm0.21}$ | $84.18_{\pm0.32}$ | $\mathbf{74.91}_{\pm0.72}$ |

In addition to the experiments on asymmetric uniform off-diagonal label noise in Tab. 1, we perform additional experiments on a different type of asymmetric label noise. We employ the label noise model

proposed by Patrini et al. Patrini et al. (2017), where for CIFAR-10: TRUCK → AUTOMOBILE, BIRD → AIRPLANE, DEER → HORSE, and CAT ↔ DOG. Differently, for CIFAR-100, the classes are grouped into 20 super-classes and within these super-classes, the labels of each class are converted into labels of the next class (circularly) with a certain probability. For this asymmetric label noise model, the test accuracy is reported in Tab. 10. We compare the test accuracies for the objective functions that have an APL-like formulation and for other methods that only propose objective functions[2], without using refined training strategies or complex architectures. The acronyms in Tab. 10, are the following: Reverse Cross Entropy (RCE), Focal Loss (FL), Asymmetric Generalized Cross Entropy (AGCE), Asymmetric Unhinged Loss (AUL), and Asymmetric Exponential Loss (AEL) (the last three have been proposed in Zhou et al. (2021)). For CIFAR-100, in Tab. 10, we used the same change of variable proposed in Novello & Tonello Novello & Tonello (2024). As for Tab. 8, $f$-NPL performs better than existing APL-like objective functions.

Table 10: Test accuracy achieved on CIFAR-10 and CIFAR-100 with asymmetric noise. An 8-layer CNN is used for CIFAR-10. The ResNet34 is used for CIFAR-100.

| Method | CIFAR-10 | | | CIFAR-100 | | |
|---|---|---|---|---|---|---|
| | 20% | 30% | 40% | 20% | 30% | 40% |
| CE | $83.00_{\pm 0.33}$ | $78.15_{\pm 0.17}$ | $73.69_{\pm 0.20}$ | $58.25_{\pm 1.00}$ | $50.30_{\pm 0.19}$ | $41.53_{\pm 0.34}$ |
| MAE | $79.63_{\pm 0.74}$ | $67.35_{\pm 3.41}$ | $57.36_{\pm 2.37}$ | $6.19_{\pm 0.42}$ | $5.82_{\pm 0.96}$ | $3.96_{\pm 0.35}$ |
| GCE | $85.55_{\pm 0.24}$ | $79.32_{\pm 0.52}$ | $72.83_{\pm 0.17}$ | $59.06_{\pm 0.46}$ | $53.88_{\pm 0.96}$ | $41.51_{\pm 0.52}$ |
| SCE | $86.22_{\pm 0.44}$ | $80.20_{\pm 0.20}$ | $74.01_{\pm 0.52}$ | $57.78_{\pm 0.83}$ | $50.15_{\pm 0.12}$ | $41.33_{\pm 0.86}$ |
| NLNL | $84.74_{\pm 0.08}$ | $81.26_{\pm 0.43}$ | $76.97_{\pm 0.52}$ | $50.19_{\pm 0.56}$ | $42.81_{\pm 1.13}$ | $35.10_{\pm 0.20}$ |
| NCE+RCE | $88.36_{\pm 0.13}$ | $84.84_{\pm 0.16}$ | $77.75_{\pm 0.37}$ | $62.77_{\pm 0.53}$ | $55.62_{\pm 0.56}$ | $42.46_{\pm 0.42}$ |
| NCE+AGCE | $88.48_{\pm 0.09}$ | $84.79_{\pm 0.15}$ | $78.60_{\pm 0.41}$ | $64.05_{\pm 0.25}$ | $56.36_{\pm 0.59}$ | $44.90_{\pm 0.62}$ |
| ANL-CE | $89.13_{\pm 0.11}$ | $85.52_{\pm 0.24}$ | $77.63_{\pm 0.31}$ | $66.27_{\pm 0.19}$ | $59.76_{\pm 0.34}$ | $45.41_{\pm 0.68}$ |
| ANL-FL | $89.09_{\pm 0.31}$ | $85.81_{\pm 0.23}$ | $77.73_{\pm 0.31}$ | $66.26_{\pm 0.44}$ | $59.68_{\pm 0.86}$ | $46.65_{\pm 0.04}$ |
| SL-NPL | $\mathbf{89.14}_{\pm 0.12}$ | $\mathbf{86.67}_{\pm 0.27}$ | $63.12_{\pm 0.48}$ | $70.90_{\pm 39}$ | $67.36_{\pm 0.74}$ | $64.59_{\pm 0.98}$ |
| GAN-NPL | $89.02_{\pm 0.10}$ | $86.14_{\pm 0.21}$ | $\mathbf{82.15}_{\pm 0.34}$ | $\mathbf{73.58}_{\pm 0.41}$ | $\mathbf{69.80}_{\pm 0.92}$ | $\mathbf{65.93}_{\pm 0.95}$ |

From Tab. 11, $f$-PNL performs better than other APL-like losses in additional realistic label noise scenarios.

Table 11: Test accuracy achieved on CIFAR-10N and CIFAR-100N, using an 8-layer CNN for CIFAR-10N, and a ResNet34 for CIFAR-100N.

| Method | CIFAR-10N | | | | | CIFAR-100N |
|---|---|---|---|---|---|---|
| | Aggregate | Random 1 | Random 2 | Random 3 | Worst | Noisy |
| CE | $85.09_{\pm 0.30}$ | $79.09_{\pm 0.28}$ | $78.59_{\pm 0.42}$ | $78.39_{\pm 0.50}$ | $61.43_{\pm 0.52}$ | $48.63_{\pm 0.53}$ |
| GCE | $87.38_{\pm 0.07}$ | $85.87_{\pm 0.27}$ | $85.43_{\pm 0.13}$ | $85.51_{\pm 0.15}$ | $75.19_{\pm 0.23}$ | $50.97_{\pm 0.60}$ |
| SCE | $88.48_{\pm 0.26}$ | $85.65_{\pm 0.30}$ | $85.71_{\pm 0.19}$ | $85.87_{\pm 0.13}$ | $73.65_{\pm 0.29}$ | $48.52_{\pm 0.11}$ |
| NCE+RCE | $89.17_{\pm 0.28}$ | $87.62_{\pm 0.34}$ | $87.66_{\pm 0.12}$ | $87.70_{\pm 0.18}$ | $79.74_{\pm 0.09}$ | $54.27_{\pm 0.09}$ |
| NCE+AGCE | $89.27_{\pm 0.28}$ | $87.92_{\pm 0.02}$ | $87.61_{\pm 0.20}$ | $87.62_{\pm 0.16}$ | $79.91_{\pm 0.37}$ | $55.96_{\pm 0.20}$ |
| ANL-CE | $89.66_{\pm 0.12}$ | $88.68_{\pm 0.13}$ | $88.19_{\pm 0.08}$ | $88.24_{\pm 0.15}$ | $80.23_{\pm 0.28}$ | $56.37_{\pm 0.42}$ |
| **SL-NPL** | $90.27_{\pm 0.34}$ | $\mathbf{88.79}_{\pm 0.37}$ | $88.71_{\pm 0.21}$ | $\mathbf{88.45}_{\pm 0.23}$ | $81.50_{\pm 0.38}$ | $52.06_{\pm 0.57}$ |
| **GAN-NPL** | $\mathbf{90.44}_{\pm 0.31}$ | $88.42_{\pm 0.29}$ | $\mathbf{88.85}_{\pm 0.33}$ | $88.34_{\pm 0.15}$ | $\mathbf{81.68}_{\pm 0.44}$ | $\mathbf{56.41}_{\pm 0.32}$ |

### D.2.3 USAGE OF $f$-NPL WITH ADVANCED TRAINING STRATEGIES

We combine the robust objective functions with refined training strategies to demonstrate that $f$-NPL can also be used in combination with complex architectures and training strategies to achieve a competitive performance with state-of-the-art techniques.

*Preliminaries.* Elaborated training strategies are frequently used to improve the performance of classification algorithms in the presence of label noise. Many techniques use ensemble models. MentorNet Jiang et al. (2018) supervises a student network by providing it a data-driven curriculum. Co-teaching Han et al. (2018) trains two networks simultaneously using the most confident predictions of one network to train the other one. For Co-teaching+ Yu et al. (2019), the authors propose to bridge

---

[2]The result of ANLs was obtained by including an L1 regularization loss in the objective function

the Co-teaching and update with disagreement frameworks. Some techniques rely on semi-supervised learning and sample selection techniques. In Berthelot et al. (2019), the authors unify many semi-supervised learning approaches in one algorithm. Divide-Mix Li et al. (2020) uses label co-refinement and label co-guessing during the semi-supervised learning phase. In Wang et al. (2022), the authors propose an algorithm that uses a new progressive selection technique to select clean samples. Shifted Gaussian Noise (SGN) Englesson & Azizpour (2024) provides a method combining loss reweighting and label correction. Contrastive frameworks have also been used in popular approaches. For instance, Joint training with Co-Regularization (JoCoR) Wei et al. (2020) aims to reduce the diversity of two networks during training, minimizing a contrastive loss. Other contrastive learning-based algorithms are proposed in Ghosh & Lan (2021); Yi et al. (2022). Other techniques rely on gradient clipping Menon et al. (2020), logit clipping Wei et al. (2023), label smoothing Wei et al. (2022), regularization Cheng et al. (2021); Liu et al. (2020; 2022); Cheng et al. (2023), meta-learning Li et al. (2019), area under the margin statistic Pleiss et al. (2020), data ambiguation Lienen & Hüllermeier (2024), thresholding Menon et al. (2015), early stopping Huang et al. (2023); Yuan et al. (2024), and joint optimization of network parameters and data labels Tanaka et al. (2018).

Table 12: Test accuracy achieved on CIFAR-10N and CIFAR-100N.

| Method | CIFAR-10N | | | | | | CIFAR-100N | |
|---|---|---|---|---|---|---|---|---|
| | Clean | Aggregate | Random 1 | Random 2 | Random 3 | Worst | Clean | Noisy |
| CE | $92.92_{\pm0.11}$ | $87.77_{\pm0.38}$ | $85.02_{\pm0.65}$ | $86.46_{\pm1.79}$ | $85.16_{\pm0.61}$ | $77.69_{\pm1.55}$ | $76.70_{\pm0.74}$ | $55.50_{\pm0.66}$ |
| FL | $93.02_{\pm0.12}$ | $88.24_{\pm0.22}$ | $86.88_{\pm0.50}$ | $86.14_{\pm0.24}$ | $87.04_{\pm0.35}$ | $79.79_{\pm0.46}$ | $76.18_{\pm0.37}$ | $57.01_{\pm1.03}$ |
| GCE | $92.83_{\pm0.16}$ | $87.85_{\pm0.70}$ | $87.61_{\pm0.28}$ | $87.70_{\pm0.56}$ | $87.58_{\pm0.29}$ | $80.66_{\pm0.35}$ | $76.35_{\pm0.48}$ | $56.73_{\pm0.30}$ |
| Co-teaching+ | $92.41_{\pm0.20}$ | $90.61_{\pm0.22}$ | $89.70_{\pm0.27}$ | $89.47_{\pm0.18}$ | $89.54_{\pm0.22}$ | $83.26_{\pm0.17}$ | $70.99_{\pm0.22}$ | $57.88_{\pm0.24}$ |
| ELR+ | $95.39_{\pm0.05}$ | $94.83_{\pm0.10}$ | $94.43_{\pm0.41}$ | $94.20_{\pm0.24}$ | $94.34_{\pm0.22}$ | $91.09_{\pm1.60}$ | $78.57_{\pm0.12}$ | $66.72_{\pm0.07}$ |
| Peer Loss | $93.99_{\pm0.13}$ | $90.75_{\pm0.25}$ | $89.06_{\pm0.11}$ | $88.76_{\pm0.19}$ | $88.57_{\pm0.09}$ | $82.00_{\pm0.60}$ | $74.67_{\pm0.36}$ | $57.59_{\pm0.61}$ |
| NCE+RCE | $90.94_{\pm0.01}$ | $89.17_{\pm0.28}$ | $87.62_{\pm0.34}$ | $87.66_{\pm0.12}$ | $87.70_{\pm0.18}$ | $79.74_{\pm0.09}$ | $68.22_{\pm0.28}$ | $54.27_{\pm0.09}$ |
| F-Div | $94.88_{\pm0.12}$ | $91.64_{\pm0.34}$ | $89.70_{\pm0.40}$ | $89.79_{\pm0.12}$ | $89.55_{\pm0.49}$ | $82.53_{\pm0.52}$ | $76.14_{\pm0.36}$ | $57.10_{\pm0.65}$ |
| Divide-Mix | $95.37_{\pm0.14}$ | $95.01_{\pm0.71}$ | $95.16_{\pm0.19}$ | $95.23_{\pm0.07}$ | $95.21_{\pm0.14}$ | $92.56_{\pm0.42}$ | $76.94_{\pm0.22}$ | $71.13_{\pm0.48}$ |
| Negative-LS | $94.92_{\pm0.25}$ | $91.97_{\pm0.46}$ | $90.29_{\pm0.32}$ | $90.37_{\pm0.12}$ | $90.13_{\pm0.19}$ | $82.99_{\pm0.36}$ | $77.06_{\pm0.73}$ | $58.59_{\pm0.98}$ |
| JoCoR | $93.40_{\pm0.24}$ | $91.44_{\pm0.05}$ | $90.30_{\pm0.20}$ | $90.21_{\pm0.19}$ | $90.11_{\pm0.21}$ | $83.37_{\pm0.30}$ | $74.07_{\pm0.33}$ | $59.97_{\pm0.24}$ |
| SOP+ | $96.38_{\pm0.31}$ | $95.61_{\pm0.13}$ | $95.28_{\pm0.13}$ | $95.31_{\pm0.10}$ | $95.39_{\pm0.11}$ | $93.24_{\pm0.21}$ | $78.91_{\pm0.43}$ | $67.81_{\pm0.23}$ |
| ProMix | $97.04_{\pm0.15}$ | $97.65_{\pm0.19}$ | $97.39_{\pm0.16}$ | $\mathbf{97.55}_{\pm0.12}$ | $\mathbf{97.52}_{\pm0.09}$ | $96.34_{\pm0.23}$ | $81.46_{\pm0.30}$ | $73.79_{\pm0.28}$ |
| ANL-CE | $91.66_{\pm0.04}$ | $89.66_{\pm0.12}$ | $88.68_{\pm0.13}$ | $88.19_{\pm0.08}$ | $88.24_{\pm0.15}$ | $80.23_{\pm0.28}$ | $70.68_{\pm0.23}$ | $56.37_{\pm0.42}$ |
| RDA | $94.09_{\pm0.19}$ | $90.43_{\pm0.03}$ | $90.09_{\pm0.29}$ | $90.40_{\pm0.01}$ | $91.71_{\pm0.38}$ | $82.91_{\pm0.83}$ | $76.21_{\pm0.64}$ | $59.22_{\pm0.26}$ |
| SGN | $94.12_{\pm0.22}$ | $92.06_{\pm0.12}$ | $91.94_{\pm0.19}$ | $91.69_{\pm0.22}$ | $91.91_{\pm0.10}$ | $86.67_{\pm0.42}$ | $73.88_{\pm0.34}$ | $60.36_{\pm0.71}$ |
| SL-NPL$_{Pro}$ | $96.08_{\pm0.20}$ | $97.19_{\pm0.16}$ | $97.00_{\pm0.17}$ | $96.93_{\pm0.09}$ | $97.07_{\pm0.12}$ | $95.34_{\pm0.35}$ | $\mathbf{82.25}_{\pm0.45}$ | $72.45_{\pm0.36}$ |
| GAN-NPL$_{Pro}$ | $\mathbf{97.20}_{\pm0.11}$ | $\mathbf{97.69}_{\pm0.21}$ | $\mathbf{97.51}_{\pm0.15}$ | $97.25_{\pm0.20}$ | $97.30_{\pm0.13}$ | $\mathbf{96.38}_{\pm0.28}$ | $81.27_{\pm0.34}$ | $\mathbf{73.93}_{\pm0.29}$ |

Since the objective function design is independent from the network architecture and training strategy, we further test $f$-NPL by integrating it with the ProMix architecture and training strategy (referring to it as $f$-NPL$_{Pro}$), maintaining the architecture and hyperparameters to the values originally proposed in Wang et al. (2022). It is important to note that these hyperparameters configurations were optimized for CE and are likely suboptimal for the various $f$-divergences employed by $f$-NPL.

The purpose of this evaluation is to show the versatility of $f$-NPL as a replacement for the CE (or other objective functions) to train state-of-the-art architectures. For instance, by implementing $f$-NPL as a class of objective functions (or simply GAN-NPL), it is possible to flexibly leverage a complex architecture and a refined training strategy based on the resources' availability. When the available hardware is powerful enough, training can be performed using the full capacity of a complex architecture and the multiple steps required by a refined training strategy. Conversely, even with limited resources, $f$-NPL can be effectively applied with a simpler architecture and a standard training strategy, still yielding satisfactory results. The same does not hold true for the CE, which has already been proven to achieve inadequate performance in the presence of label noise. We evaluate $f$-NPL$_{Pro}$ under realistic label noise in Tab. 12. Despite using ProMix's original hyperparameters, which are optimal for the CE and have not been tuned for other $f$-divergences, $f$-NPL$_{Pro}$ achieves top-tier performance across different scenarios. These empirical findings demonstrate the effectiveness of $f$-NPL$_{Pro}$, showing that by combining $f$-NPL and complex architectures and training strategies, it is possible to attain performance comparable to state-of-the-art approaches.

### D.2.4 COMPUTATIONAL COMPLEXITY

In this section, we compare the computational complexity of CE and $f$-NPL, including the proposed correction approaches. First, we compare the computational complexity of CE and $f$-NPL without

correction. The objective function of $f$-NPL comprises two terms: the first term corresponds to the output neuron of $D$ (neural network) corresponding to the sample's label; the second term is the summation of the output neurons of $f^*(D)$, where, for the considered $f$-divergences, $f^*(\cdot)$ is available in closed-form. Therefore, after the forward pass, the first term of the objective of $f$-NPL requires the evaluation of the network output corresponding to the label, thus having similar time requirements to the CE. The second term of the objective of $f$-NPL requires computing the summation of a function of the network outputs, which has similar time requirements to the computation of the first term. However, this additional complexity is negligible w.r.t. the complexity of the forward and backward passes. In summary, $f$-NPL has approximately the same computational complexity as the CE, as we show in Tab. 13 (using the hardware described in D.1). Since the posterior correction is performed during the test phase, the computational complexity of the training process remains unchanged. In this case, during the test phase, there is only an additional summation of two vectors, which is negligible w.r.t. the computational complexity of the entire forward pass.

The small computational complexity is actually a great advantage of the posterior correction approach. Regarding the objective function correction approach, instead, the computational complexity slightly increases for the estimate of the bias, which requires a significantly smaller amount of time compared to forward and backward passes (see Tab. 13). Finally, for mini WebVision, the average time required for $f$-NPL and CE coincides: 4m:25s (4 minutes and 25 seconds).

Table 13: Comparison of computational complexity (in seconds). The quantity measured (Meas. type) is of three types: "Loss" refers to the time for the loss computation, "Train Epoch" refers to the time for the whole training epoch, and "Test Epoch" refers to the time required for the test epoch. Each value is computed as the average over 50 measurements.

| Dataset | Meas. type | CE | $f$-NPL | $f$-NPL$_p$ | $f$-NPL$_o$ |
|---------|-----------|-----|---------|-------------|-------------|
| CIFAR-10 | Loss | $50 \cdot 10^{-6}$ | $240 \cdot 10^{-6}$ | $240 \cdot 10^{-6}$ | $3.3 \cdot 10^{-6}$ |
| | **Train Epoch** | **22.5** | **22.5** | **22.5** | 23.0 |
| | **Test Epoch** | **1.9** | **1.9** | **1.9** | **1.9** |
| CIFAR-100 | Loss | $50 \cdot 10^{-6}$ | $280 \cdot 10^{-6}$ | $280 \cdot 10^{-6}$ | $41 \cdot 10^{-6}$ |
| | **Train Epoch** | **34.5** | **34.5** | **34.5** | 53.67 |
| | **Test Epoch** | **2.7** | **2.7** | **2.7** | **2.7** |

