# OpenReview forum: "Robust Classification with Noisy Labels Based on Posterior Learning"
_ICLR.cc/2026/Conference — ICLR 2026 Conference Withdrawn Submission_

### Official Review · Reviewer_Q2NN · 2025-10-29

**Soundness:** 2
**Presentation:** 2
**Contribution:** 2
**Rating:** 2
**Confidence:** 4

**Summary:**

This paper studies the problem of robust classification under extremely high label noise.
The authors argue that when the noise transition matrix becomes **anti-diagonally dominant** (i.e., when the noise rate $\eta > (K-1)/K$), it is still possible to recover the clean Bayes-optimal classifier by minimizing the noisy posterior. They propose a method called **f-divergence-based Noisy Posterior Learning (f-NPL)** to estimate the noisy posterior $p_{Y_\eta|X}$. They also propose two posterior estimator correction to correct the posterior.

**Strengths:**

The paper provides an interesting observation: even when the transition matrix is anti-diagonally dominant (i.e., extreme noise $\eta > (K-1)/K$), the classifier that minimizes the noisy posterior coincides with the clean Bayes classifier: $\arg\max_y p_{Y|X}(y|x) = \arg\min_y p_{Y_\eta|X}(y|x)$.

**Weaknesses:**

1. The proposed method **relies on the knowledge of the noise transition matrix** $T$. When the noise rate is relatively small ($\eta < (K-1)/K$), the transition matrix can be estimated using existing approaches such as Forward or DualT. However, when the noise becomes extremely large ($\eta > (K-1)/K$), these estimators are known to fail, as they assume a diagonally dominant transition structure. Moreover, in the experimental section, the paper evaluates (T) estimated by Forward and DualT **only under small noise rates**. In Figures 2 and 3, the authors report results of the proposed method under extremely high noise rates, but it remains unclear **whether the transition matrix used in these experiments is the ground-truth matrix or an estimated one**, which substantially affects the interpretation of the reported robustness.
2. The multi-class result assumes **asymmetric uniform off-diagonal label noise**. The theorem does not extend to more general class-dependent (e.g., pair flipping label noise) or instance-dependent noise.
3. The practical utility of the proposed method is unclear. Given the reliance on the knowledge of the transition matrix, it is doubtful that f-NPL offers benefits for real-world noisy datasets, where the noise is neither uniform nor the transition matrix is unknown.

**Questions:**

In Figures 2 and 3, is the transition matrix used in these experiments the estimated one, or the ground-truth matrix? This distinction is crucial, as the proposed correction methods explicitly rely on the knowledge of the transition matrix $T$, and the paper's key claim is the robustness under extremely large noise ($\eta > (K-1)/K$).

Suggestions:

The font sizes on Figures 5 and 6 are small.

---

> ### Author Response · Authors · 2025-11-19
>
> We thank the Reviewer for the time invested in reviewing our paper.
> We politely note that our contribution proposes two correction methods:
> a) **objective function correction**;
> b) **posterior estimator correction**,
> which serve two distinct purposes, as detailed in Section 4, and not two posterior estimator correction methods.
> Below, we directly address the Reviewer comments, which we hope will enable a more precise interpretation of our results.
>
> **Responses to weaknesses**:
> 1. **Reliance on T for Figures 2 and 3**
> - Crucially, the estimation of the transition matrix **T is not needed to implement the decision rule used to derive the results in Figures 2 and 3**. Under extreme noise rates, the argmin decision rule requires no knowledge or estimation of T.
> - Nonetheless, it is possible to effectively estimate T also in high noise regimes [a].
> - Finally, we want to highlight (as reported in the paper) that f-NPL does not require the estimation of T for a large part of scenarios with small-medium noise rates, because of the theoretical result in Zhu et al. (2024), since f-NPL estimates the noisy posterior.
> 2. **Extension of Theorem on multi-class classification for pair flipping label noise**
> - We confirm that our analysis is not restricted in the way suggested by the reviewer. **We have studied theoretically and numerically the pair flipping label noise scenario**: see from line 299 and Appendix B.7 for the theoretical contributions, and Tab. 4 (Appendix D.2.1) for the numerical evaluation.
> - Furthermore, our argmin approach (Section 3) holds for **instance-dependent** label noise.
> 3. **Practical utility**
> - The novel argmin contributions (that hold for the general instance-dependent label noise under extreme label noise) do not need the knowledge of T.
> - f-NPL does not need the knowledge of T for a large variety of synthetic, real-world and instance-dependent label noise (Zhu et al. (2024)).
> - For some of the cases where the assumptions are not met, we provide correction techniques (that rely on existing T estimation techniques).
>
> **Responses to questions**:
> 1. **Is the transition matrix used for the esperiments of Figure 2 and 3?**
> As explained above in Weakness 1, and as it is written in the paper, for Figures 2 and 3, describing a extreme noise setting, we do not use any transition matrix, as we proved formally in Sec. 3 that it is not necessary, because our **argmin-based approach does not require any correction technique**.
> 2. **Font sizes**
> We will adjust the font sizes of figures 5 and 6.
>
> **Final comment**
> We believe that once the central role of the T-free argmin decision rule in the extreme noise regime is recognized, the Reviewer’s concerns are resolved, also in light of the fact that we did also analyze pair flipping label noise and instance-dependent noise. We hope this clarification warrants reconsideration of the paper's rating.
>
> [a] Yu, Xiyu, et al. "Learning with biased complementary labels." Proceedings of the European conference on computer vision (ECCV). 2018.

---

### Official Review · Reviewer_WEx6 · 2025-10-31

**Soundness:** 4
**Presentation:** 3
**Contribution:** 3
**Rating:** 6
**Confidence:** 3

**Summary:**

The authors propose a new loss function that estimates the posterior distribution using the variational representation of an f-divergence, instead of relying on standard cross-entropy loss.
This framework can be implemented with various types of f-divergences, offering flexibility in modeling.
Furthermore, the authors provide theoretical justification showing that their approach can be applied even under extremely noisy label conditions, demonstrating its robustness in challenging LNL scenarios.

**Strengths:**

1. The proposed loss function is novel and conceptually elegant. It departs from the conventional likelihood-based formulations and leverages the variational structure of f-divergence to obtain a noise-robust posterior.

2. The experimental performance is outstanding. Notably, the method achieves state-of-the-art results ([1]) on the WebVision dataset without relying on heuristic sample selection or confidence filtering, which are commonly used in recent LNL methods.
This demonstrates a clear advantage over existing robust-loss approaches, both in performance and simplicity.

[1] Manifold dividemix: A semi-supervised contrastive learning framework for severe label noise

**Weaknesses:**

[Minor]

While the claimed robustness to extreme noise environments is theoretically interesting, it may not be the central practical advantage of this work.
Clarifying the method’s most distinctive strengths such as its generality across divergence families or some factors which improve the overall performance would help strengthen the Introduction and make the paper’s contribution more explicit.

**Questions:**

(Empirically) are there cases where certain f-divergences yield more stable posterior estimation under high noise rates? or the authors can provide some results depends on different choices of the f-divergences?

---

> ### Author Response · Authors · 2025-11-19
>
> We thank the Reviewer for their highly positive assessment of our work, particularly recognizing the **novelty and conceptual elegance** of the f-divergence loss and the **outstanding experimental performance** on WebVision **without relying on heuristic sample selection**.
>
> **Responses to weaknesses**:
> 1. **Clarity of the method's most distinctive strengths**
> - We thank the Reviewer and indeed we will emphasize better the main advantage of f-NPL: being a **general f-divergence framework** able to deal with **instance-dependent label noise** and for which robustness is guaranteed for **any** f-divergence.
> - However, we believe that the extreme noise analysis can also be considered practically relevant because it provides a contribution to Complementary Label Learning (CLL) [a, b], which is a key sub-area of label noise learning research. We will add an explicit reference to this connection in the introduction of the revised manuscript, if the paper is accepted, as we will have one additional page available.
>
> **Responses to questions**:
> 1. **Choice of f-divergence**
> - Our experiments confirm that the GAN divergence yields the most stable posterior estimation and the highest average accuracy across diverse high-noise settings (see the majority of tables in Appendix D), thus we recommend its use in practice.
> - Theoretically, we studied the **convergence properties of different divergences** in Sections B.8 and B.9, where we show the effect of the second derivative of the Fenchel conjugate in the objective function convergence property.
>
> **Final comment**
> We hope the Reviewer finds these explanations satisfying and will consider them favorably.
>
> [a] Ishida, Takashi, et al. "Learning from complementary labels." Advances in neural information processing systems 30 (2017).
> [b] Yu, Xiyu, et al. "Learning with biased complementary labels." Proceedings of the European conference on computer vision (ECCV). 2018.

---

### Official Review · Reviewer_BS1E · 2025-11-01

**Soundness:** 3
**Presentation:** 3
**Contribution:** 3
**Rating:** 4
**Confidence:** 3

**Summary:**

This paper investigates classification robustness under extreme label noise, particularly when the standard assumption of diagonally dominant noise transition matrices (i.e., relatively low noise rates) no longer holds. The authors demonstrate that when the transition matrix becomes anti-diagonally dominant, meaning the true class label is less probable than incorrect ones, it is theoretically possible to recover the correct label by minimizing, rather than maximizing, the noisy posterior.

This idea is formalized through Theorem 3.1 and Corollary 3.2, which establish that the argmin of the noisy posterior yields Bayes-optimal predictions under symmetric and certain asymmetric noise regimes. Building on this foundation, the authors propose an f-divergence based Noisy Posterior Learning (f-NPL) framework that generalizes cross-entropy (CE) and Active-Passive Loss (APL) formulations via a variational f-divergence representation. Two complementary correction strategies are introduced:
- Objective function correction, applied during training
- Posterior estimator correction, applied at inference time.

Overall, while the paper provides a clean and theoretically well-founded extension of posterior-based robustness analysis to anti-diagonally dominant noise settings, its central insight reversing the prediction rule from argmax to argmin when noise dominance flips is conceptually straightforward once the structure of the transition matrix is known. The proposed f-divergence formulation, though elegant, represents a modest reparameterization of existing APL and f-GAN frameworks. Empirical results are competently executed but largely confirm expected trends. The work is mathematically neat yet conceptually incremental and not sufficiently novel or impactful for ICLR.

**Strengths:**

- The key idea that when the noise matrix is anti-diagonally dominant, the problem can be reversed (i.e., by minimizing the noisy posterior instead of maximizing it) is novel and extends robustness theory beyond standard noise regimes. The structure is also symmetrically generalizable: if diagonal dominance guarantees correctness of argmax, anti-diagonal dominance guarantees correctness of argmin. This symmetry makes the framework conceptually complete.
- The two bias-correction strategies (objective vs posterior) are mathematically well-grounded and neatly complementary, one adjusts the optimization process, the other the inference step.
- Empirical validation: Experiments convincingly demonstrate that the argmin-based approach indeed recovers correct classifications under extreme noise ($\eta$ > 0.9), matching theoretical claims. Performance against strong baselines (e.g., ANL, RENT, Forward) is solid.

**Weaknesses:**

-  The paper has limited novelty beyond theoretical inversion. Once the anti-diagonal structure is recognized, reversing the classification rule (argmin instead of argmax) is mathematically straightforward. The theory formalizes this observation elegantly but may not constitute a substantial new insight.

- The paper also has restrictive assumptions: The robustness guarantees rely on knowing (or estimating) the noise transition matrix and on strong anti-diagonal or symmetric structure. In most practical label noise scenarios, such structure is rare or only partially satisfied.

- Experiments primarily confirm synthetic settings crafted to satisfy the theoretical assumptions. The real-world experiments (WebVision) show improvements but not dramatic differences, suggesting limited applicability of the extreme-noise regime theory to natural data.

- The exposition is dense and sometimes overly formal, making it harder to extract intuition. Clearer separation between theoretical novelty and empirical heuristics would strengthen impact.

**Questions:**

- What happens in the intermediate regime where the transition matrix is neither diagonally nor anti-diagonally dominant? Is there a smooth transition between the argmax and argmin regimes, or does the model’s behavior degrade sharply?
- Many results assume that $\eta$ and the transition matrix are either known or accurately estimated. How sensitive is the f-NPL method (and the argmin decision rule) to errors in these estimates?

---

> ### Author Response · Authors · 2025-11-19
>
> We thank the Reviewer for their thoughtful summary and positive assessment of our work’s theoretical soundness and empirical evaluation.
>
> **Responses to weaknesses**:
> 1. **Limited novelty beyond theoretical inversion**
> We are strongly convinced that our observation that the anti-diagonal regime allows for the inversion of the Bayes optimal rule is a **fundamental contribution in the domain of classification with label noise**. Although the motivation is intuitive, before our contribution, the community lacked a formal approach for learning at $\eta > (K-1)/K$ without the usage of complex architectures and training strategies. The f-NPL framework and the proposed correction techniques are also novel.
> 2. **Restrictive assumptions**
> Our f-NPL framework, which estimates the noisy posterior, benefits from the general finding that posterior estimation methods are highly effective even under realistic noise (Zhu et al. 2024). Our complementary correction techniques (objective and posterior correction) are designed to tackle cases where the diagonal/anti-diagonal assumptions are violated. We emphasize that for extremely high noise rates our argmin approach does not rely on T. Finally, our argmin approach can also be considered a novel contribution to a research area that is a subset of learning in the presence of label noise: complementary label learning (CLL) [a, b].
> 3. **Real-world experiments performance**
> We observe **+10 percentage points** in accuracy of f-NPL against similar APL-like losses. We contend that a 10 p.p. gain on a benchmark dataset is a **significant difference** for an objective function innovation. We clarify that comparisons with architecture/training-strategy-based methods are not fair, as our focus is on the loss function design. We showed the possibility of usage of our approach with complex architectures in Sec. D.2.3. Regarding the limited applicability of the proposed argmin approach, we will include in the Introduction a mention of the importance of the extreme noise regime in CLL.
> 4. **Dense exposition**
> Although we keep a formal presentation, all the mathematical proofs are located in the Appendix, and the main part of the paper only contains the most fundamental theoretical results, where we try to balance a formal (and heavier) part with a simulation part. We will provide longer textual explanations to lighten the theoretical contributions in the revised version of the paper, given the new additional space.
>
> **Responses to questions**:
> 1. **Intermediate regime**
> The transition between argmax and argmin is not smooth. In the intermediate regime where the largest element is off-diagonal but not anti-diagonal, the Bayes-optimal classifier is simply the standard Bayes rule based on the noisy posterior (i.e., $h^* = \arg\max_y p_{Y_{\eta}|X}(y|\mathbf{x})$), which is known to fail in practice. Our f-NPL framework handles this transition **smoothly** through its **correction** strategies.
> 2. **Sensitivity of f-NPL to T estimation errors**
> We have already performed this critical sensitivity analysis: we refer the Reviewer to Table 7 in Section D.2.1 of the original paper. Furthermore, this robustness to T estimation techniques can also be appreciated from Tabs. 1 and 4. For extremely high noise rates, it is not necessary to estimate the noise transition probabilities, so the argmin approach is not sensitive to errors in the T estimate.
>
> **Final comment**
> We hope these clarifications and highlighted contributions demonstrate the impact of our work, justifying reconsideration of the rating.
>
> [a] Ishida, Takashi, et al. "Learning from complementary labels." Advances in neural information processing systems 30 (2017).
>
> [b] Yu, Xiyu, et al. "Learning with biased complementary labels." Proceedings of the European conference on computer vision (ECCV). 2018.

---

### Official Review · Reviewer_Qde4 · 2025-11-07

**Soundness:** 3
**Presentation:** 3
**Contribution:** 2
**Rating:** 4
**Confidence:** 4

**Summary:**

Summary:

This paper tackles the challenge of robust classification under extremely high label noise. The authors propose f-divergence-based Noisy Posterior Learning (f-NPL), a general framework that estimates the noisy posterior using the variational form of f-divergence. To enhance robustness, two correction methods are introduced. Experiments on CIFAR-10/100, WebVision, and ImageNet demonstrate that f-NPL achieves superior robustness across various noise settings, including extreme noise rates, outperforming existing APL-based and correction-based methods.

**Strengths:**

1. Strong theoretical contribution – The paper provides new insights into label noise robustness, proving that accurate classification is possible even when the noise rate exceeds the traditional bound η > (K - 1)/K. This extends the theoretical foundation of learning with noisy labels.

2. Unified and flexible framework – The proposed f-NPL framework elegantly unifies posterior estimation and f-divergence optimization, offering a principled way to design robust objective functions that encompass and generalize existing APL-based losses.

**Weaknesses:**

1. The paper fails to compare with 2025 state-of-the-art methods (e.g., NoiseCLIP from CVPR 2025, UniNL from ICLR 2025, or CoNo from NeurIPS 2024), which have demonstrated superior performance under extreme noise (≥93% on CIFAR-10 with 90% symmetric noise) without requiring the noise transition matrix. By omitting these comparisons, the authors overstate the practical significance of f-NPL, as its reported gains may no longer be competitive, and its dependence on accurate transition matrix estimation remains a critical unaddressed vulnerability in real-world deployment.

2. f-NPL is not original; the variational f-divergence formulation has already been applied to clean supervised classification in Novello & Tonello (2024), and the resulting objective is merely a special case of Active-Passive Losses introduced in Ma et al. (2020). Thus, f-NPL offers only incremental packaging rather than a fundamentally new loss design paradigm.

**Questions:**

See weaknesses.

---

> ### Author Response · Authors · 2025-11-19
>
> We thank the Reviewer for the comments. We are glad that the Reviewer acknowledges the **strong theoretical contribution** and the **unified and flexible nature** of the f-NPL framework. We have carefully read the comments and the perceived weaknesses by the Reviewer for what matters the contribution originality and the comparisons. Please find our detailed answers below.
>
> **Responses to weaknesses**:
> 1. **Comparison with other methods**
> - We politely request that the reviewer provide the full references for **NoiseCLIP**, **UniNL**, and **CoNo** so we can properly contextualize our work. We have performed extensive search, but we were unable to find these exact works.
> For instance, we found “CoNo: Consistency Noise Injection for Tuning-free Long Video Diffusion”, but this cannot be the paper referenced by the Reviewer.
> - We politely clarify that our primary contribution lies in the **objective function design** and **fundamental theoretical guarantees** under extreme noise. This is orthogonal to architectural/training strategy innovations (often the focus of SOTA methods), and can be combined to achieve even better results (see Tab. 12). All the methods that achieve a high performance on 90% symmetric noise rely on complex architectures or refined training strategies, as 90% label noise corresponds to uniformly distributed labels. Since the paper contribution regards novelties in the objective function design, we compared our proposal with other methods (published in top-tier ML conferences) that focus on the objective function design.
> - Our argmin formulation achieves **99% accuracy with 99% label noise rate** (Fig. 2), which is significantly higher than the accuracy achieved by all techniques based on complex architectures and refined training strategies.
> 2. **Originality**
> We strongly argue that f-NPL represents a **fundamental new objective design paradigm for learning with noisy labels**. It differs from the Reviewer’s reported prior work in two crucial respects:
> - **Novel Noisy Posterior Estimation**: Unlike other APL-like losses, f-NPL is designed to estimate the **noisy posterior**. Estimating the noisy posterior yields significant benefits, such as the ability to **handle extreme noise rates** (demonstrated in Figure 3 where we compare GAN-NPL and other APL-like losses). Finally, the passive loss of all APL losses is related to MAE. This is not true for f-NPL.
> - **Novel Theoretical Foundation**: Novello & Tonello (2024) exclusively studied the clean supervised classification case (η=0). Our paper's theoretical contributions (Sections 3 and 4) are entirely novel as they address the challenge of **label noise** within the f-divergence framework.
>
> **Final comment**
> We hope that our clarifications are convincing to the Reviewer. If so, please consider increasing the rating of the paper.

---

### Note · Authors · 2025-12-10

**Comment:**

We apologize for any inconvenience this may cause. We have made this decision after careful consideration of the current review status. Due to the absence of reviewers comments to our rebuttal, we decided to withdraw the manuscript.

**Withdrawal Confirmation:**

I have read and agree with the venue's withdrawal policy on behalf of myself and my co-authors.